# Unlocking the Power of Function Vectors for Characterizing and Mitigating Catastrophic Forgetting in Continual Instruction Tuning

**Gangwei Jiang**[1,2], **Caigao Jiang**[4], **Zhaoyi Li**[1,2], **Siqiao Xue**[4], **Jun Zhou**[4],
**Linqi Song**[2]*, **Defu Lian**[1]*, **Ying Wei**[3]*
[1] University of Science and Technology of China,    [2] City University of Hongkong ,
[3] Zhejiang University,    [4] Independent
{gwjiang, lizhaoyi777}@mail.ustc.edu.cn, linqi.song@cityu.edu.hk
liandefu@ustc.edu.cn, ying.wei@zju.edu.cn

## Abstract

Catastrophic forgetting (CF) poses a significant challenge in machine learning, where a model forgets previously learned information upon learning new tasks. Despite the advanced capabilities of Large Language Models (LLMs), they continue to face challenges with CF during continual learning. The majority of existing research focuses on analyzing forgetting patterns through a single training sequence, thereby overlooking the intricate effects that diverse tasks have on model behavior. Our study explores CF across various settings, discovering that model forgetting is influenced by both the specific training tasks and the models themselves. To this end, we interpret forgetting by examining the function vector (FV), a compact representation of functions in LLMs, which offers a model-dependent indicator for the occurrence of CF. Through theoretical and empirical analyses, we demonstrate that CF in LLMs primarily stems from biases in function activation rather than the overwriting of task processing functions. Leveraging these insights, we propose a novel function vector guided training methodology, incorporating a regularization technique to stabilize the FV and mitigate forgetting. Empirical tests on four benchmarks confirm the effectiveness of our proposed training method, substantiating our theoretical framework concerning CF and model function dynamics. Our code is available at https://github.com/GangweiJiang/FvForgetting.git.

## 1 Introduction

Continual instruction tuning (Peng et al., 2023; Chung et al., 2024) has emerged as an indispensable ingredient in the development of Large Language Models (LLMs) (Brown et al., 2020; Radford et al., 2019; Touvron et al., 2023b), enabling them to meet the demands of specific domains (Roziere et al., 2023; Thirunavukarasu et al., 2023; Xue et al., 2024) and human preferences (Ouyang et al., 2022). However, a notable concern with such continual tuning is "catastrophic forgetting" (McCloskey & Cohen, 1989; Kirkpatrick et al., 2017), where models may lose essential skills (Dou et al., 2023; Chen et al., 2023) such as mathematical reasoning while adjusting to user instructions. While instruction tuning effectively evolves LLMs, it's critical to characterize and mitigate forgetting within these models.

Research on LLM forgetting (Luo et al., 2024; Wang et al., 2023c; Wu et al., 2024a) generally examines alterations in specific abilities like reading comprehension, factual knowledge, mathematical reasoning skills, and so on, underscoring the universal existence of catastrophic forgetting. As they have primarily studied from a single training sequence, they fail to establish the connection between model forgetting and the characteristics of training data. Concurrently, there is a notable gap in understanding the internal mechanisms that underlie model forgetting. To date, only a limited body of research has ventured into this area; notably, the work of Kotha et al. (2024), proposing the task

---

*Corresponding authors

inference hypothesis, explores how conflicts between task processors lead to forgetting. Nevertheless, the existing literature still struggles to track the internal mechanisms behind forgetting, which is crucial for understanding why and when forgetting occurs in language models after learning new tasks and how to avoid it.

In this study, we conduct thorough experiments on various continual instruction tuning benchmarks covering multiple language models, task sequences, and evaluation metrics. Our investigation focuses on the research question: When does forgetting happen? The experimental results suggest that model forgetting is a complex outcome of various factors, including the nature of training and test tasks, and the state of the model. However, traditional methodologies for evaluating task similarities to characterize forgetting—such as those based on feature similarity (Ramasesh et al., 2020; Lee et al., 2021) and readout similarity (Lee et al., 2021)—tend to overlook the distinctive task-related information inherent in different models. Meanwhile, purely model-dependent measurements, like the L2 distance of model parameters after being fitted on a new task (Lin et al., 2023; Evron et al., 2024), necessitate training to compute task similarity. We identify a critical gap in the availability of robust tools to dissect and understand the processes underlying forgetting.

To this end, we utilize the Function Vector approach (Todd et al., 2023), a method grounded in mechanistic interpretability (Wang et al., 2023a; Bills et al., 2023), which represents the input-output function within a model into a compact vector form. Our analysis begins with a revisitation of the theoretical formulation of FV through the perspective of latent variable models (Baum & Petrie, 1966; Gruber et al., 2007), and establishes that the FV serves as a traceable latent variable in LLMs. We then examine model forgetting through the Function Vector perspective, successfully identifying occurrence of forgetting by evaluating the similarity between training and testing tasks (Sec. 3). Subsequent empirical investigations lead us to conclude that the fundamental driver of forgetting is the shift in the mapping from the input $x$ to the latent concept variable $\theta$, rather than erasure of previously learned task functions (Sec. 4).

Based on our analysis, we conclude that minimizing the shift in the function vector during training serves as a key strategy for mitigating forgetting. We propose a function vector guided training mechanism as a simple yet efficient design for mitigating forgetting. This approach involves limiting changes to the function vectors associated with training tasks through a regularization term, coupled with the adoption of a function vector-guided Kullback-Leibler (KL) divergence loss. This loss function aims to diminish the discrepancies between logits derived from zero-shot input and those adjusted by function vector intervention, ensuring the fine-tuned model remains consistent with the inner task function. Validated across multiple datasets and models, this method significantly alleviates forgetting in both general and in-context learning abilities, confirming the correlation between FV dynamics and forgetting.

**Main findings and contributions.** (1) We investigate catastrophic forgetting in LLMs covering multiple language models, task sequences, and evaluation metrics, discovering that forgetting in LLMs is highly model-dependent, and assert a new analytical tool for characterizing forgetting behavior. (2) Using empirical and theoretical analysis based on the function vector framework, we reveal that forgetting generally results from the activation of biased model functions rather than overwriting previous functions. (3) We develop a function vector guided training approach that preserves and aligns function vectors during fine-tuning, significantly improving both general and in-context learning across multiple continual learning datasets.

## 2 PRELIMINARIES

### 2.1 CATASTROPHIC FORGETTING

Continual learning (Serra et al., 2018; Wu et al., 2024c;b) seeks to tackle the core challenge of incrementally learning from a sequence of real-world tasks over time, specifically addressing how to adapt to new tasks without forgetting previously learned knowledge – a phenomenon widely known as catastrophic forgetting (McCloskey & Cohen, 1989; Kirkpatrick et al., 2017).

In this paper, we focus on continual learning of a language model $M_0$, which has already been pre-trained on a vast data corpus $\mathcal{D}_{PT}$ using language modeling tasks (Brown et al., 2020; Radford et al., 2019) followed by preference optimization via human feedback (Ouyang et al., 2022). Specif-

ically, we assume a stream of tasks $T_1, T_2, \ldots, T_N$, where each $j$-th task $T_j$ consists of a dataset $\mathcal{D}_j = \{x_j^i, y_j^i\}$, with $x_j^i$ and $y_j^i$ representing the inputs and outputs text sequences, respectively. On each task $T_j$, the model $M_{j-1}$ is optimized towards minimization of the loss $\mathcal{L}_{T_j}(M_{j-1})$, coupled with a continual learning objective if applied, resulting in the updated model $M_j$. This continual learning process, applied into a language model, is commonly referred to as continual instruction tuning (Peng et al., 2023; Chung et al., 2024).

## 2.2 FUNCTION VECTOR

Following the mechanistic interpretability work in LLMs (Todd et al., 2023; Hendel et al., 2023), we investigate the internal workings of a task on LLMs through function vector — compact vector representation of input-output task identified within the hidden states of transformers during in-context learning (ICL (Brown et al., 2020)). An activation patching (Meng et al., 2022; 2023; Wang et al., 2023a) procedure is performed on the ICL hidden states to determine the casual set of attention heads that mediate tasks. These heads collaborate to convey a function vector (FV) which represents the LLM's function specific to input-output task. Function vector is regarded as an efficient way to characterize function execution within LLMs (Todd et al., 2023).

Formally, for a given dataset $\mathcal{D}^T$ of task $T$, the function vector $\theta_T$ is derived through two steps:

First, the activation patching is performed to determine the attention heads set (denoted as $\mathcal{S}$) with significant cause-and-effect relationship between ICL inputs and correct outputs. Specifically, the model will run on a counterfactual ICL input $[\hat{p}, x]$ incorporating a label-shuffled prompt $\hat{p} = [(x_1, \hat{y}_1), \ldots, (x_n, \hat{y}_n)]$, which typically leads to incorrect outcomes. Then the representation at specific head for $[\hat{p}, x]$ is substituted by the averaged task activation $\bar{h}_{lk}^T$ and calculated its causal effect (CE) on the model's output.

$$\text{CE}_{lk}([\hat{p}, x]) = P_{M^{h_{lk}^T \rightarrow \bar{h}_{lk}^T}}(y \mid [\hat{p}, x]) - P_M(y \mid [\hat{p}, x]). \tag{1}$$

Here, $\bar{h}_{lk}^T \in \mathbb{R}^d$ symbolizes the mean activations of in-context learning input for task $T$ at the last token position across layer $l$ and head $k$, with $d$ being the dimension of the layer output as $h_{lk}^T = head_{lj} W_{lj}^O$ is the equivalent of the attention head output in the residual stream (Elhage et al., 2021). $M^{h_{lk}^T \rightarrow \bar{h}_{lk}^T}$ denotes the model with a replacement operation on attention head $(l, k)$ at last token. A higher CE implies that the specific head's state is critical for accurate predictions which encodes of more task-relevant information. In this paper, $\mathcal{S}$ is the attention heads with top-10 CE.

Second, function vector $\theta_T$ is assembled by summing up the averaged ICL inputs activation from the attention heads within $\mathcal{S}$, formulated as $\theta_T = \sum_{(l,k) \in \mathcal{S}} \bar{h}_{lk}^T \in \mathbb{R}^d$. The comprehensive methodology for this extraction process can be found in the Appendix F.

In this paper, we revisit the definition of FV and study its dynamics before and after learning a new task, providing a surrogate lens to uncover the inherent mechanisms of forgetting in LLMs.

## 3 CATASTROPHIC FORGETTING OF LLMS

This section empirically examines the research question of *when forgetting occurs during continual instruction tuning*, specifically in relation to *task types*, *training stages* and *language models*.

**Datasets.** We build the task sequences for continual instruction tuning using SuperNI (Wang et al., 2022), a collection of NLP tasks with expert-written instructions that are unseen to the pre-trained model. SuperNI is commonly used for assessing cross-task generation and conflict after fine-tuning language models. To investigate the relationship between forgetting and task types, i.e., generation and classification, we design six task sequences. These sequences consist of *pure generation tasks*, *pure classification tasks* and *mixed sequences containing both generation and classification tasks*. Their main information is listed in Table 4, with additional details available in Appendix D.

**Evaluation metrics.** We adopt the following five metrics to quantify various aspects of forgetting: (1) **GP** $= \frac{1}{N^g} \sum_{j=1}^{N^g} a_N^{T_j^e}$, which is the average zero-shot performance across $N^g$ general evaluation tasks after instruction tuning on the final $N$-th task. Here, $a_m^q$ denotes the zero-shot performance on task $q$ after sequentially tuning the $m$-th task, and $T_j^e$ refers to the $j$-th general evaluation task. Performance is uniformly measured using Rouge-L (Lin, 2004), where classification

Table 1: Final performance on 3 SuperNI benchmarks on 4 language models. Hella., Com., Alpa., and Ob. denote evaluation score on Hellswag, CommonsenseQA, Alpaca, Object Count datasets, respectively. The $\Delta$ value in red bold style is compared to performance of their initial model $M_0$. Higher **Forget** or lower $\Delta$ represents more forgetting. ***Main conclusion:*** *Forgetting consistently occurs in both general and newly learned tasks, showing considerable variations depending on the types of tasks, stages of training, and the specific language models involved.*

| | Zero-Shot Performance in General Task | | | | | In-Context Performance in General Task | | | | | Performance in Trained Task | | |
|---|---|---|---|---|---|---|---|---|---|---|---|---|---|
| | Hella. | Com. | Alpa. | Ob. | GP ↑/Δ ↑ | Hella. | Com. | Alpa. | Ob. | IP ↑/Δ ↑ | AP ↑ | FP ↑ | Forget ↓ |
| **Llama2-7b-chat** | | | | | | | | | | | | | |
| $M_0$ | 57.89 | 57.37 | 26.5 | 27.12 | **42.22** | 58.95 | 57.89 | 35.17 | 34.21 | **46.55** | / | / | / |
| NI-Seq-C1 | 47.37 | 40.00 | 32.00 | 31.61 | **-4.48** | 24.21 | 27.89 | 28.89 | 26.84 | **-19.60** | 86.10 | 83.80 | **2.30** |
| NI-Seq-G1 | 48.95 | 39.47 | 27.36 | 39.72 | **-3.35** | 37.89 | 42.63 | 28.84 | 38.95 | **-9.48** | 24.96 | 19.35 | **5.61** |
| NI-Seq-M1 | 52.11 | 42.63 | 31.09 | 29.51 | **-3.39** | 45.79 | 31.05 | 24.58 | 33.16 | **-12.91** | 59.02 | 54.32 | **4.69** |
| **Llama3-8b-chat** | | | | | | | | | | | | | |
| $M_0$ | 81.58 | 58.42 | 22.64 | 40.04 | **50.67** | 85.26 | 63.16 | 27.42 | 49.47 | **56.32** | / | / | / |
| NI-Seq-C1 | 79.47 | 46.84 | 23.27 | 32.32 | **-5.20** | 79.47 | 40.00 | 25.62 | 45.79 | **-8.60** | 83.40 | 82.10 | **1.30** |
| NI-Seq-G1 | 72.63 | 35.79 | 22.05 | 29.39 | **−10.70** | 67.89 | 31.05 | 19.77 | 41.58 | **-16.26** | 28.29 | 21.09 | **7.19** |
| NI-Seq-M1 | 78.42 | 40.00 | 21.93 | 21.58 | **-10.19** | 76.84 | 40.00 | 21.32 | 35.91 | **-12.81** | 60.74 | 52.62 | **8.11** |
| **Mistral-7b-instruct** | | | | | | | | | | | | | |
| $M_0$ | 73.68 | 60 | 24.74 | 5.02 | **40.86** | 79.47 | 66.32 | 32.36 | 37.89 | **54.01** | / | / | / |
| NI-Seq-C1 | 63.16 | 50.00 | 32.04 | 15.3 | **-0.75** | 66.84 | 51.05 | 36.8 | 37.89 | **-5.87** | 84.70 | 85.40 | **-0.70** |
| NI-Seq-G1 | 57.37 | 45.26 | 26.3 | 13.81 | **-5.18** | 57.89 | 35.79 | 32.04 | 39.47 | **-12.71** | 27.62 | 19.77 | **7.85** |
| NI-Seq-M1 | 65.26 | 47.89 | 33.02 | 12.35 | **-1.23** | 63.68 | 38.42 | 34.79 | 45.79 | **-8.34** | 61.96 | 57.00 | **4.95** |
| **Llama2-13b-chat** | | | | | | | | | | | | | |
| $M_0$ | 69.47 | 51.05 | 28.99 | 15.09 | **41.15** | 75.26 | 57.89 | 35.46 | 43.16 | **52.94** | / | / | / |
| NI-Seq-C1 | 65.79 | 52.63 | 34.18 | 21.51 | **+2.38** | 66.32 | 48.42 | 38.48 | 38.95 | **-4.90** | 83.20 | 82.26 | **0.93** |
| NI-Seq-G1 | 63.16 | 38.95 | 28.12 | 13.84 | **-5.13** | 65.79 | 32.11 | 30.92 | 34.21 | **-12.18** | 25.64 | 18.16 | **7.47** |
| NI-Seq-M1 | 71.58 | 49.47 | 34.10 | 28.09 | **+4.66** | 70.53 | 48.42 | 36.51 | 37.37 | **-4.73** | 60.10 | 56.33 | **3.76** |

accuracy equals Rouge-L with output post-processing (Zhao et al., 2024). We set $N^g = 4$ general evaluation tasks, covering Hellaswag (Zellers et al., 2019), CommonsenseQA (Talmor et al., 2018), Alpaca (Hendrycks et al., 2020), and BBH-Object-Count (Srivastava et al., 2022), and this setting also applies to 4 and is extended to $N^g = 7$ in Sec. 6. (2) $\mathbf{IP} = \frac{1}{N^g} \sum_{j=1}^{N^g} \hat{a}_N^{T_j^e}$, which is the average in-context performance on $N^g$ general evaluation tasks after tuning on the last $N$-th task. The in-context performance, $\hat{a}_m^q$, is conditioned on the $n$-shot ICL input $[p, x]$, where $p = [(x_1, y_1), ..., (x_n, y_n)]$ with $n = 5$ for this work. (3) $\mathbf{FP} = \frac{1}{N} \sum_{j=1}^{N} a_N^{T_j}$, which is the average zero-shot performance across all $N$ instruction tuning tasks after tuning on the final $N$-th task. $T_j$ represents the $j$-th instruction tuning task in the sequence. (4) $\mathbf{AP} = \frac{1}{N} \sum_{j=1}^{N} a_j^{T_j}$, which is the average zero-shot performance when learning every $j$-th instruction tuning task. (5) $\mathbf{Forget} = \mathbf{AP} - \mathbf{FP}$, which has been widely adopted Wu et al. (2022); Ke et al. (2023) to measure forgetting. More detailed information about the datasets and evaluation metrics is presented in Appendix D.

**Instruction tuning setup.** For each task in the sequence, we continually fine-tune four language models, including Llama2-7b-chat, Llama2-13B-chat (Touvron et al., 2023b), Llama3-8B-chat (Dubey et al., 2024), Mistral-7B-instruct-v2.0 (Jiang et al., 2023) using the causal language model loss (Radford et al., 2019). Unless otherwise noted, we use the LoRA fine-tuning approach (Hu et al., 2021), employing the Adam optimizer with a learning rate of $1e^{-4}$ and a batch size of 32. Additional implementation details can be found in the Appendix E.

We report the results on continual learning of four language models on three SuperNI sequences in Table 1, highlighting that forgetting occurs across general ability, in-context learning ability, and fine-tuned ability. Higher scores for general and in-context abilities signify better mitigation of forgetting. For fine-tuned ability, we report the values of **AP**, **FP** and their difference **Forget**, where a lower **Forget** value indicates less forgetting. A detailed breakdown of the results for the other three SuperNI sequences is provided in Appendix F. Figure 1 illustrates the forgetting of the Llama2-7b and Llama3-8b models during continual learning on NI-seq-C1 and NI-seq-G1. The vertical axis signifies the training stage, where M$i$ denotes the model after completing the $i$-th task. The horizontal axis displays test performance across tasks, with the performance of $M0$ (without continual instruction tuning) shown at the top. The heatmaps show the relative shifts in performance compared to $M0$, with declines indicated by bluer values and improvements by redder colors.

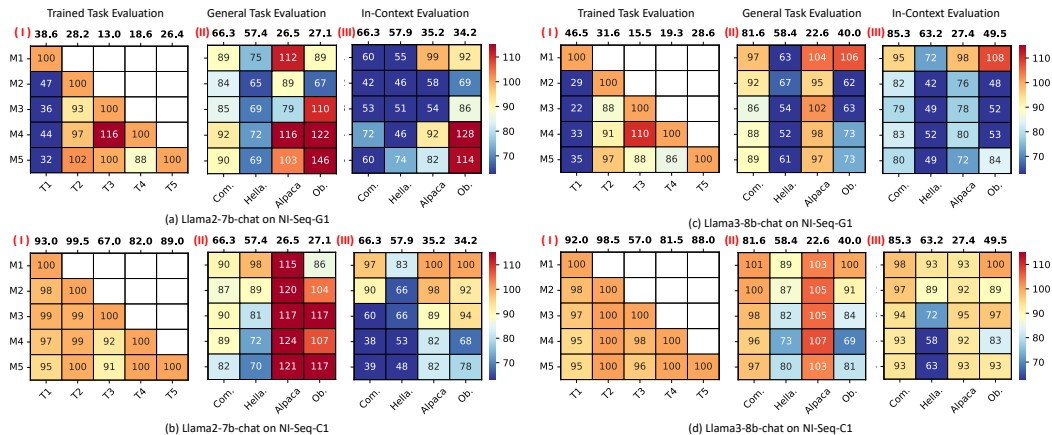

Figure 1: Performance heatmap during continual learning of 2 different sequences on Llama2-7b-chat and Llama3-8b-chat. The numbers above the heatmap indicate the baseline performance of each task, with the performance of the pre-trained model for general testing (e.g., in a-(II) 66.3 is the score of Commonsense on original Llama2-7b-chat) and performance right after completing current task for trained task testing (e.g., in a-(I) 28.2 is the score of T2 on Llama2-7b-chat post 2-th task training). The numbers on the heatmap show the percentage change relative to the baseline (e.g., in a-(I) the first column 47 indicates the score at 38.6*47%). ***Main conclusion: (1) Learning generation tasks (a/c) vs. classification tasks (b/d) leads to more forgetting. (2) Forgetting may reduce naturally (a-(II)/d-(II)); (3) Forgetting is model-dependent (a/b vs. c/d).***

From Table 1 and Fig. 1, we observe two major trends: (1) consistent forgetting of LLMs across both general and newly acquired tasks, irrespective of the task sequence type, and (2) considerable variability in forgetting across evaluation tasks – e.g., Hellaswag appeals more prone to forgetting than Alpaca. Beyond these general findings, we further investigate how forgetting is influenced by task types, training stages, and the specific language models employed.

**Task type perspective: generation task sequences lead to greater forgetting.** Upon analyzing the performance of NI-Seq-C1 (classification tasks) and NI-Seq-G1 (generation tasks) in Table 1, we observe a noticeable increase in forgetting when learning generation task sequences. For instance, the Llama3-8b-chat model shows larger performance declines with NI-Seq-G1 (10.7, 16.2, and 7.2 in general, in-context, and trained tasks, respectively) compared to NI-Seq-C1 (5.2, 8.6, and 1.3 declines). Additionally, the forgetting score **Forget** of all models over all tasks in the NI-Seq-C1 sequence is below 2.3, suggesting a stronger ability to mitigate forgetting for LLMs than small language models (Qin & Joty, 2022; Razdaibiedina et al., 2023). We further evaluate LLMs with more sequences, obtaining similar observations as shown in Appendix F.

**Training stage perspective: forgetting may naturally mitigate during training.** Figure 1 illustrates the extent of forgetting at each training phase across various evaluation datasets. Contrary to the nearly consistent performance drop seen in previous studies (Luo et al., 2024; Wu et al., 2022), we frequently observe a phenomenon where performance initially decreases but later rebounds. For instance, the fourth column of **(a)-(II)** shows the "Ob." test score dropping to 67% on the M2 model; however, after two stages, the performance leaps to 122%. This contradiction further raises questions about how sequential fine-tuning on new tasks impacts the internal capabilities of LLMs, allowing them to recover previously forgotten capabilities.

**Model perspective: forgetting is model-dependent.** We compare forgetting between LLMs, specifically, for the "Ob." task performance shown in **(a)-(II)** and **(c)-(II)**, continual instruction tuning of Llama2-7b-chat demonstrates a performance increase of up to 146% relative to using the Llama2-7b-chat itself, while that of Llama3-7b-chat shows a decrease to 73%. This suggests that forgetting is not only task-related but also heavily influenced by model-dependent factors such as model size, architecture, and the diversity of pre-training data. These factors shape each model's unique capacity for tasks, revealing that the mere feature similarity between tasks (e.g., hidden states in the last layer) is insufficient to predict model-dependent forgetting patterns (see Appendix F).

## 4 CORRELATIONS BETWEEN FUNCTION VECTORS AND FORGETTING

The previous section prompts a more effective measure for characterizing catastrophic forgetting, surpassing those traditionally used in continual learning research with small models, such as feature similarity (Ramasesh et al., 2020; Lee et al., 2021) and readout similarity (Lee et al., 2021) between tasks. They fail to overlook the distinctive task-related information inherent in different models. Other model-dependent measures, such as the $\ell2$-distance of model parameters after tuning on new tasks (Lin et al., 2023; Evron et al., 2024). In addition, we have proven that these traditional measures (e.g., feature similarity and $\ell2$-distance) are loosely correlated with forgetting in LLMs in Fig. 6. In this section, we first establish that *the similarity between function vectors (FV, see Sec. 2) is strongly correlated with diverse forgetting patterns across task types, training stages, and language models.*

**Forgetting coincides with changes in FV similarity between model updates.** We now explore the relationship between forgetting and variations in function vectors across evaluation tasks. As shown in Figure 2, we evaluate the performance on the four general evaluation tasks throughout the sequence for both generation and classification settings, alongside shifts in function vectors $\theta_{T^e}$. "Fv sim" in the diagram refers to $\mathrm{Cosine}(\theta_{T^e}^0, \theta_{T^e}^j)$, where $\theta_{T^e}^j$ is the FV of evaluation task $T^e$ after fine-tuning the $j$-th task. We observe a clear consistency between the performance decline and variations in function vectors. Specifically, as performance drops, the similarity between FV $\theta_{T^e}^0$ and FV $\theta_{T^e}^j$ generally declines, whereas this similarity tends to increase as performance recovers. For example, in the Hellaswag task within NI-Seq-G1 (top right subplot in Figure 2), the correlation coefficient ($R^2$ value) between zero-shot performance and our proposed similarity measure reaches 0.873. This finding underscores that fluctuations in the FV often coincide with model forgetting, and justifies the rationale of characterizing forgetting through function vectors.

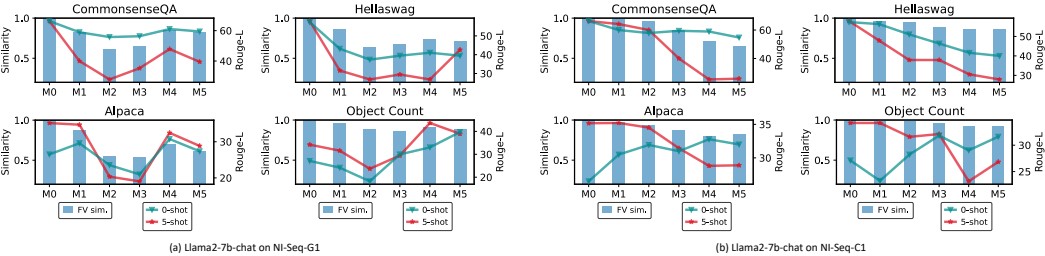

Figure 2: The shifts in function vector with 0/5-shot performance during tuning. The bar chart corresponding to the left y-axis shows the similarity of function vectors to their initial state. The line graph corresponding to the right y-axis depicts the model's Rouge-L metric on test data. ***Main conclusion:*** *There exists a significant correlation between performance (line data) and FV similarity (bar data).* The correlation plots with more data point are provided in Fig. 6.

## 5 CAUSAL PATHWAY TO FORGETTING THROUGH FUNCTION VECTORS

Before delving into how function vectors causally influence forgetting, we revisit the function vector $\theta_T$ from the perspective of latent variable models (Baum & Petrie, 1966; Gruber et al., 2007). The specific FV $\theta_T$ in Sec. 2 is derived from the sum of activations of certain attention heads in the LLM via causal analysis, when provided with ICL input of task $T$. $\theta_T$ represents a latent variable descriptive of task $T$; conditioning predictions on $\theta_T$ produces a model function specific to task $T$, i.e.,

$$P_M(y|x, \theta_T = \sum_{(l,k)\in\mathcal{S}} h_{lk}) \to f_T(y|x). \tag{2}$$

Here, $f_T$ characterizes the function of task $T$ within the model, where different function vectors activate distinct functions, enabling the model to exhibit diverse abilities.

Prior research (Xie et al., 2021; Wang et al., 2024a) has established that under a latent variable assumption, in-context learning in LLMs can be rewritten as:

$$P_M\left(y \mid x_1^T, y_1^T, \ldots, x_n^T, y_n^T, x\right) = \int_{\Theta} P_M(y \mid \theta, x) P_M\left(\theta \mid x_1^T, y_1^T, \ldots, x_n^T, y_n^T, x\right) d\theta, \tag{3}$$

where $P_M$ denotes the predictive probability of the LLM $M$, and $\theta$ is a potentially high dimensional latent variable within the space $\Theta$. For example, in the task of predicting the antonym ($y$) of a word ($x$), the latent variable could correspond to "writing the antonym of the word" ($\theta$). This framework indicates that ICL boosts performance by deducing the correct $\theta$ from observed input-output pairs.

By comparing this framework with Eq. 2, we conclude that LLM predictions hinge on two critical factors: first, *the prediction given by the model's task-specific function* $P_M(y|x,\theta_T)$, and second, *the capacity of an input $x$ to accurately trigger its corresponding Function Vector* $\theta_T$ (i.e., $P_M(\theta_T|x)$). Previous work in continual learning has considered the role of task-specific functions (*a.k.a,* model parameters) on forgetting, sparked methods such as gradient orthogonalization (Lopez-Paz & Ranzato, 2017; Chaudhry et al., 2018) and parameter regularization (Kirkpatrick et al., 2017; Wu et al., 2024c) to mitigate its impact. However, our research hypothesizes that in continual instruction tuning of LLMs, the *intrinsic cause of forgetting is the bias of the latent variable* $\theta_T$ *elicited by the input $x$*, as supported by the following empirical findings.

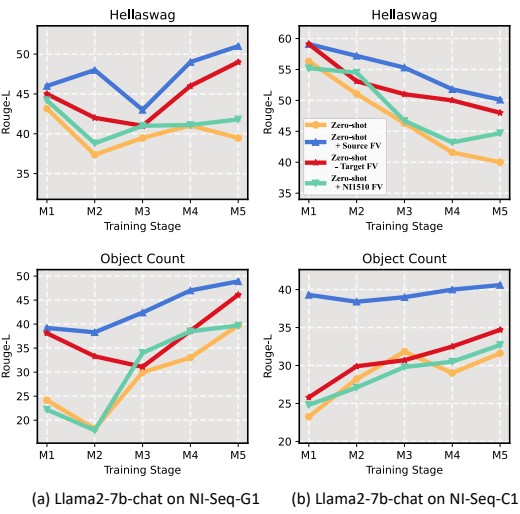

Figure 3: Intervention results on fine-tuned model. '+ Source FV' and '- Target FV' refers to Evidence I and Evidence II, respectively. ***Main conclusion:** intervention with related function vector mitigating forgetting.*

**Finding I: forgetting is mitigated by adding the source FV with respect to the evaluation task.** For each evaluation task $T^e$, we conducted an intervention experiment during continual instruction tuning by adding the source function vector, i.e., $\theta_{T^e}^0$, into the LLMs (blue line in Figure 3). Here, the source FV $\theta_{T^e}^0$ is extracted from the original LLM $M_0$ following the procedure described in Section 2. The intervention involves adding the function vector to a specific layer during inference ($h_l = h_l + \theta_{T^e}^0$), and the reported results reflect the best performance obtained by iterating over layers [3, 6, 9, 12, 15]. The findings indicate that integrating the source FV $\theta_{T^e}^0$ into inference – which explicitly transforms $P_M(y|x)$ to $P_M(y|x,\theta_{T^e}^0)$ – substantially restores model performance. For instance, Figure 3(a) shows an average performance increase of 7.3 on Hellaswag through this intervention. This improvement signifies that explicitly injecting the latent variable $\theta_{T^e}^0$ helps identify the function specific to the evaluation task $T^e$, i.e.,$P_M(y|x,\theta_{T^e}^0)$, thereby mitigating forgetting.

**Finding II: forgetting is mitigated by subtracting the biased FV with respect to the current training task.** Similarly, we conduct the intervention experiment again on each evaluation task $T^e$ by subtracting the target FV, i.e., $\theta_{T_j}^j$, as depicted by the red line in Figure 3. Here, the target FV $\theta_{T_j}^j$ represents the function of task $T_j$ after the model has been trained on $j$-th task. It is calculated as $\sum_{(l,k)\in\mathcal{S}} \bar{h}_{lk}^{T_j,j}$, where $\bar{h}^{T_j,j}$ denotes the mean activation from in-context learning inputs in task $T_j$. The intervention involves modifying the representations in a specific layer during inference through subtracting the function vector ($h_l = h_l - \theta_{T_j}^j$). The results reveal a surprising reduction in forgetting. For example, removing the target FV during Hellaswag inference, as shown in Fig. 3 (b) top, increases the model M5's performance from 39.5 to 49.1. This finding suggests that the groundtruth latent variable $\theta_{T^e}^0$ has been confounded by the biased target FV introduced during training on incoming tasks; substracting it mitigates this interference and thus reduces forgetting.

**Finding III: forgetting of the in-context learning ability is severe.** In Table 1 and Fig. 1, one of the most significant observations is that the forgetting of in-context learning (ICL) ability is notably more severe than the zero-shot ability when evaluated on general tasks. For instance, in Llama2-7b-chat trained on NI-Seq-C1, the forgetting of Hellaswag reaches 34.7 for ICL, while in the zero-shot setting, the value is just 8.94. We attribute this to the overfitting of the mapping $P_M\left(\theta \mid x_1^T, y_1^T, \ldots, x_n^T, y_n^T, x\right)$ to a biased target FV in task $T_j$. This results in a distorted, complex mapping as evident from the correlation between CommonsenseQA 5-shot evaluations and FV shifts

during NI-Seq-G1 training on Llama2-7b-chat (Figure 2). Notably, 5-shot performance declines faster than zero-shot as the FV shifts from M2 to M3, but recovers when it stabilizes at M5.

To summarize, the above empirical findings, alongside the strong correlations between forgetting and changes in FVs demonstrated in Section 4, and our causal mediation analysis in Appendix F which reveals a shift in the set $\mathcal{S}$ of causal attention heads (i.e., signifying the magnitude and direction of the function vector) during continual instruction tuning, collectively advocate that: (1) forgetting is primarily driven by modifications in $P_M(\theta|x)$, rather than changes in $P_M(y|x,\theta)$; (2) in LLMs, the function responsible for handling a specific task is not overwritten but is instead overshadowed by newly introduced functions.

## 6 FUNCTION VECTOR GUIDED TRAINING DESIGN

In this section, we present the function vector guided training design that serves as an effective mechanism for mitigating the forgetting of general abilities applicable to a wide range of language models and continual learning baselines. We introduce the overall architecture, present experimental results, and analyze how function vector guided training works.

**Function vector guided training.** The correlation between forgetting and the function vector implies a principle for training method design. That is, training should be capable of maintaining the correct mapping of inputs to function vectors and thus mitigating forgetting. Based on this principle, we propose function vector-guided training as a simple yet efficient design to mitigate forgetting. Our method introduces two novel regularization terms:

First, to mitigate the introduction of a biased function vector during training, we restrict the alterations in FVs tied to the training tasks, effectively maintaining the model's $P_M(\theta_T|x)$ unchanged. To this end, restrictions are imposed on the activation values of specific heads signifying the function vector with a FV consistency loss. When training task $T_j$, the loss is specified as follows:

$$\ell_{FV} = \sum_{(l,k)\in\mathcal{S}} d\left(h_{lk}^{M_{j-1}}(x), h_{lk}^{M}(x)\right), \tag{4}$$

where $h_{lk}^{M}(x)$ denotes the activations on last input token of head $j$ in layer $l$ on input $x$ from model $M$ and $M_{j-1}$ is the model before training task $T_j$. $d(\cdot,\cdot)$ is the distance metrics between variables, and we adopt L2 distance in this paper.

Furthermore, we introduce a FV-guided KL-divergence loss to align the task function raised by zero-shot inputs and the FV-intervened one. The detailed function for training task $T_j$ is as below:

$$\ell_{KL} = KL[P_M(\cdot \mid x)\|P_{M_{j-1}^{h_l\rightarrow h_l+\theta_{T_j}^0}}(\cdot \mid x)]. \tag{5}$$

$P_M(\cdot \mid x)$ is the predicted probability distribution for each token in the vocabulary $\mathcal{Y}$ given input $x$. $M_{j-1}^{h_l\rightarrow h_l+\theta_{T_j}^0}$ denotes the model trained after $j-1$ tasks with intervention by the function vector $\theta_{T_j}^0$, that is $\theta_{T_j}^0$ is added to the activation $h_l$ of layer $l$ during the forward pass. $l$ is selected as 9 in this paper. Then, the overall optimization objective is to minimize the loss $\ell = \ell_{LM} + \alpha_1\ell_{FV} + \alpha_2\ell_{KL}$, where $\ell_{LM} = -\log P_M(y|x)$ is the language modeling loss (Brown et al., 2020) and $\alpha_1, \alpha_2$ are trade-off hyper-parameters. The algorithm procedure and implementation details are provided in B and E, respectively. This FV-guided fine-tuning leverages the function $P_M(y|x,\theta_T)$ within the model to guide the fine-tuning, ensuring that the model retains a robust causal pathway $P_M(y|x,\theta_T)P_M(\theta_t|x)$ after fine-tuning and minimizes the impact of newly introduced functions on past abilities.

**Main results.** The experiments were conducted on three language models, demonstrating their effectiveness in combination with existing continual learning methods, such as Incremental Lora Hu et al. (2021) (**IncLora**), Elastic Weight Consolidation Kirkpatrick et al. (2017) (**EWC**), Orthogonal Lora Wang et al. (2023b) (**OLora**), and Instruction-based Memory Replay Wang et al. (2024b) (**InsCL**). Besides the three task sequences we adopted in the previous sections, we also addressed the effectiveness of our approach on the public benchmark TRACE Wang et al. (2023c), which

Table 2: Performance of baselines and their improved version with Function Vector Guided (**FVG**) training on four benchmarks. ***Main conclusion: FVG significantly prevent forgetting in general and in-context learning capabilities (GP and IP).***

| | Method | NI-Seq-G1 GP ↑ | IP ↑ | FP ↑ | NI-Seq-C1 GP ↑ | IP ↑ | FP ↑ | NI-Seq-M1 GP ↑ | IP ↑ | FP ↑ | TRACE GP ↑ | IP | FP ↑ |
|---|---|---|---|---|---|---|---|---|---|---|---|---|---|
| Llama2-7b-chat | $M_0$ | 49.85 | 54.43 | | 49.85 | 54.43 | | 49.85 | 54.43 | | 49.85 | 54.43 | |
| | LoraInc | 47.16 | 30.94 | 19.35 | 45.83 | 27.71 | 83.80 | 47.55 | 37.23 | 54.33 | 46.17 | 38.08 | 41.20 |
| | +FVG | +3.34 | +25.25 | +2.84 | +3.98 | +25.53 | +1.70 | +2.65 | +15.78 | +3.52 | +6.92 | +16.17 | +12.13 |
| | Ewc | 33.48 | 26.87 | 17.72 | 46.08 | 38.76 | 85.00 | 44.47 | 41.69 | 55.85 | 49.07 | 47.98 | 54.22 |
| | +FVG | +15.73 | +27.18 | +0.85 | +3.11 | +15.96 | +0.37 | +6.18 | +13.99 | +0.01 | +2.21 | +6.01 | -8.77 |
| | O-lora | 45.15 | 31.90 | 22.67 | 41.54 | 20.54 | 79.33 | 50.16 | 39.52 | 56.94 | 36.96 | 29.38 | 37.13 |
| | +FVG | +4.89 | +23.59 | +0.11 | +8.38 | +33.93 | +6.2 | +0.29 | +14.95 | -0.42 | +14.32 | +24.61 | +8.32 |
| | InsCL | 45.80 | 41.79 | 27.14 | 44.03 | 35.69 | 81.67 | 49.76 | 43.09 | 60.83 | 46.46 | 41.63 | 52.95 |
| | +FVG | +2.65 | +8.30 | +0.91 | +5.00 | +16.11 | +1.23 | +0.98 | +8.32 | -2.22 | +6.70 | +11.04 | +0.92 |
| Llama3-8b-c. | $M_0$ | 56.61 | 60.61 | | 56.61 | 60.61 | | 56.61 | 60.61 | | 56.61 | 60.61 | |
| | LoraInc | 45.51 | 39.85 | 21.10 | 51.89 | 54.63 | 82.10 | 48.00 | 47.82 | 52.63 | 50.31 | 52.61 | 27.14 |
| | +FVG | +7.79 | +15.31 | +3.10 | +3.99 | +5.19 | +0.30 | +4.88 | +4.75 | +5.78 | +3.84 | +6.29 | +7.22 |
| | InsCL | 46.48 | 49.46 | 28.53 | 52.11 | 57.30 | 82.50 | 49.46 | 53.50 | 60.92 | 51.87 | 51.22 | 37.32 |
| | +FVG | +6.60 | +8.06 | -0.85 | +3.52 | +1.58 | -0.60 | +4.34 | +2.75 | -2.80 | +2.04 | +7.92 | +6.66 |
| Mistral-7b-i. | $M_0$ | 47.55 | 57.51 | | 47.55 | 57.51 | | 47.55 | 57.51 | | 47.55 | 57.51 | |
| | LoraInc | 42.81 | 38.82 | 19.78 | 48.00 | 53.00 | 85.4 | 49.79 | 51.02 | 57.01 | 51.91 | 51.37 | 44.68 |
| | +FVG | +4.49 | +16.61 | +0.64 | +2.35 | +2.67 | -0.50 | -2.41 | +4.02 | +0.43 | -1.49 | +5.14 | +10.37 |
| | InsCL | 43.46 | 51.06 | 25.78 | 40.77 | 49.49 | 83.03 | 42.38 | 52.27 | 58.01 | 50.90 | 50.39 | 55.99 |
| | +FVG | +2.71 | +4.64 | -0.30 | +6.75 | +4.27 | +2.07 | +6.13 | +3.40 | -0.84 | -0.98 | +6.12 | +1.19 |

includes a learning sequence comprised of multi-choice QA, code generation, mathematical reasoning, and summarization tasks. Table 2 shows the continual instruction tuning performance on four benchmarks, leading to several key observations:

*Observation 1*: Function vector guided training significantly reduces forgetting in both general and in-context learning capabilities. Traditional continual learning methods experience substantial forgetting, with InsCL performing somewhat better yet still experiencing a minimum decline of 11.34% on **IP**. By contrast, function vector guided training achieves consistent and substantial gains in combating this issue, enhancing performance in the Llama2-7b-chat across all baselines, achieving average increases of 5.44 in **GP** and 17.52 in **IP**. Function vector-guided training provides a methodology to break through the forgetting of general knowledge in existing models.

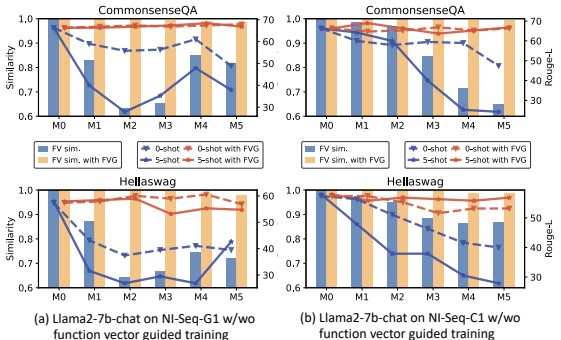

(a) Llama2-7b-chat on NI-Seq-G1 w/wo function vector guided training

(b) Llama2-7b-chat on NI-Seq-C1 w/wo function vector guided training

Figure 4: The shifts in function vector with 0/5-shot performance with function vector guided training. ***Main conclusion: FVG prevents the shift in FV (yellow bar) and thus mitigates forgetting (orange line).***

*Observation 2*: function vector guided training does not compromise the plasticity in learning new tasks. This technique demonstrates a negligible reduction in the **FP** metric, signifying a well-maintained balance between plasticity and stability. For the TRACE datasets with the IncLora method, our method shows an improvement of 12.13 on the Llama2-7b-chat model, showing large potential in protecting performance on the training sequence as well. Nonetheless, under certain scenarios, like when utilizing the InsCL replay method on NI-Seq-M1, our strategy yields a 2.8 drop in **FP**. This could be attributed to the conflict between the diverse gradient information from the memory buffer and our regularization component.

*Observation 3*: The effectiveness of continual learning methods varies with the type of training sequence. As discussed above, different training tasks impact existing capabilities in distinct ways. For instance, while learning classification tasks results in relatively minor forgetting, sequences of generative tasks tend to lead to more pronounced issues. Consequently, the conventional approach of uniformly applying the same strategy to every training task demonstrates significant variance in performance across different sequences. Specifically, InsCL exhibits a performance decline on the

NI-Seq-C1 dataset compared to IncLora, while it shows improvements on the other three datasets. In contrast, our method retains the model's task-specific functions and thoroughly accounts for the characteristics of the current learning task, thereby achieving consistent and considerable enhancements across a diverse range of datasets.

**How does function vector-guided training work?** Following our principle of designing function vector-guided training, we test the mechanics of this approach by examining the shifts in function vectors during training. We visualize the alteration of function vectors during function vector-guided training in Fig. 4, following the setting in Sec. 4. It becomes evident that the methods we propose can effectively maintain the shifts in function vectors for general tasks, even without the need to access their training data. As the FVs remain stable during the fine-tuning phase, the performance on general tasks is effectively safeguarded.

# 7 RELATED WORK

**Catastrophic forgetting in fine-tuned language models.** Fine-tuning foundational LLMs (Touvron et al., 2023a;b) has become a generic technique for enhancing their capacity of following instructions (Wei et al., 2022; Zhang et al., 2024a;b) and mastering domain-specific content (Yue et al., 2023; Christophe et al., 2024). However, adopting such technique can have a negative effect of hurting the original ability of LLMs, which is widely known as Catastrophic Forgetting (Kirkpatrick et al., 2017; Luo et al., 2024; Kotha et al., 2024; Wu et al., 2024b). In context of LLMs, existing approaches towards mitigating this issue can mostly be categorized into three types: regularizing the update of model parameters (Huang et al., 2021; Cha et al., 2021), replaying previous or self-synthesized data (Scialom et al., 2022; Huang et al., 2024a) and resisting interference via parameter-efficient fine-tuning (Razdaibiedina et al., 2023; Wang et al., 2023b). In this work, we aims to characterize CF in LLMs through the function vector, concluding that such forgetting primarily stems from biases in function activation rather than the overwriting of task processing functions. To this end, We propose function vector guided training, a regularization-based method to protect task activation from being improperly destroyed during fine-tuning to cure the forgetting issue.

**Mechanistic analysis to fine-tuning.** Existing works on analyzing the internal mechanism (Räuker et al., 2023; Ferrando et al., 2024) of fine-tuning mainly focus on how LLMs acquire new capacity, arguing that models learn a minimal transformation on top of the original capability (Jain et al., 2024) (wrappers), subtractable and reusable parameter shift vectors (Huang et al., 2024b; Gao et al., 2024) (task vectors) and to align input queries with their internal knowledge that are already acquired in the pre-training stage (Ren et al., 2024). Nevertheless the inherent reason for the forgetting issue brought by fine-tuning currently remains unclear, and hence our work instead targets on this important point. We have successfully identified the compact task representation, known as the function vector, can tracks task forgetting in LLMs. Our empirical data indicate a strong correlation between shifts in the function vector and the phenomenon of task forgetting.

# 8 CONCLUSION

In this study, we tackle the issue of catastrophic forgetting in LLMs via a detailed investigation using the FV approach, highlighting its pivotal role in characterizing and mitigating forgetting phenomena. Our analysis across a vast array of benchmarks reveals that model forgetting is intricately linked to shifts in latent concept variables (characterized by function vector), facilitated by our novel function vector-guided training strategy. This method, integrating a regularization term with a function vector-guided KL divergence loss, significantly curtails forgetting, thereby enhancing both general and in-context learning capabilities of LLMs in continual learning settings.

ACKNOWLEDGMENTS

This work was supported in part by grants from the National Natural Science Foundation of China (No. U24A20253), the Research Grants Council of the Hong Kong SAR under Grant GRF 11217823 and Collaborative Research Fund C1042-23GF, the National Natural Science Foundation of China under Grant 62371411, InnoHK initiative, the Government of the HKSAR, Laboratory for AI-Powered Financial Technologies.

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

## A    FUNCTION VECTOR EXTRACTION

We next consider how to extract $\theta_c$ for a given dataset $D^c$, drawing on the concept of function vectors proposed by Todd et al. (2023). This extraction is carried out using in-context learning (ICL) samples, where the model incorporates task-relevant information into its hidden states as it engages with examples with the ICL prompt. This process is associated with the emergence of $\theta_c$ (Todd et al., 2023; Hendel et al., 2023). Subsequently, a causal mediation analysis (Pearl, 2013; Vig et al., 2020; Li et al., 2024) is conducted on the ICL inputs to identify attention heads with significant causal impacts on the output, and aggregating their representations results in $\theta_c$. Interestingly, this vector remains effective even under zero-shot input scenarios. The detailed procedure is outlined below:

First, we start by gathering the task-conditioned activation for each model head by averaging the ICL input representation of the given task $D^c$, i.e.,

$$\bar{h}_{lj}^c = \frac{1}{|D^c|} \sum_{(x) \in D^c} h_{\ell j}([p, x]).$$

(6)

Where $p = [(x_1, y_1), ..., (x_N, y_N)]$ represents the N-shot ICL prompt text made up of held-out samples of task $c$, $h_{lj}$ is the model activation at the last token, layer $l$ and position $j$, and $\bar{h}_{lj}^c$ represents the task-conditioned activations.

Then to assess the existence of a cause-and-effect relationship between $\bar{h}_{lj}^c$ and correct output, we employ causal mediation analysis. The model will run on a counterfactual ICL input $[\hat{p}, x]$ incorporating a label-shuffled prompt $\hat{p} = [(x_1, \hat{y}_1), ..., (x_N, \hat{y}_N)]$, typically leading to incorrect outcomes. We then substitute the value of the specific head with the task-specific conditioned activation $\bar{h}_{lj}$ and calculate its causal effect (CE) on the model's output.

$$\text{CE}_{lj}([\hat{p}, x]) = P_{M^{h_{lj} \to \bar{h}_{lj}^c}}(y_i \mid [\hat{p}, x]) - P_M(y_i \mid [\hat{p}, x]).$$

(7)

Here, $M^{h_{lj} \to \bar{h}_{lj}^c}$ denotes the model with a replacement operation on attention head $(l, j)$ at last token of the input sentence. A higher CE suggests that the specific head's state is crucial in enabling accurate predictions, denoting the encoding of more task-relevant information. For each head at layer $l$ and position $j$, we adopt the approach proposed by Todd et al. (2023) to calculate the average CE across a variety of tasks. Subsequently, we identify the top 10 heads with the highest average CE (recorded as set $\mathcal{S}$) as the most critical in conveying task-relevant information. The task vector $\theta_c$ is is then obtained by aggregating the task-conditioned activation from the attention heads in the set $\mathcal{S}$, i.e., $\theta_c = \sum_{(l,j) \in \mathcal{S}} \bar{h}_{lj}^c$.

We then evaluates the effectiveness of the function vector ($\theta_c$) through intervention experiments on the initial model across multiple datasets. Results show that the FV significantly influences the output behavior for specific tasks, with its introduction notably improving zero-shot performance in certain tasks and removal diminishing the model's ability to produce correct outputs. This suggests that the model's specific abilities can be identified and analyzed by studying the corresponding FV.

## B    FUNCTION VECTOR GUIDED TRAINING ALGORITHM

Algorithm 1 outlines the procedure for function vector guided continual learning. It begins with a sequence of tasks, each paired with its corresponding dataset, as well as a pre-trained language model (referred to as $M_0$). Based on the approach from Todd et al. (2023), a collection of held-out datasets $\{\bar{D}_1, \bar{D}_2, \ldots, \bar{D}_K\}$ is utilized to determine the set of function vector heads. Furthermore, it is proposed that the function vector head set $\mathcal{S}$ is applicable across different datasets, allowing us to collect this set $\mathcal{S}$ only once.

---

**Algorithm 1:** Function vector guided training procedure

---

**Input:** Given a sequence of tasks $\{T_1, T_2, ....T_N\}$ and their corresponding datasets $\{D_1, D_2, ....D_N\}$, a series of held-out dataset $\{\bar{D}_1, \bar{D}_2, ....\bar{D}_K\}$ to figure out the set of function vector heads, pre-trained language model $M_0$

**Output:** Language model $M_N$ trained after $N$ tasks

1  **Function** `Main`($\{D_1, D_2, ....D_N\}$, $\{\bar{D}_1, \bar{D}_2, ....\bar{D}_K\}$, $M_0$)**:**

2     $\mathcal{S} \leftarrow$ `GetFunctionVectorSet`($\{\bar{D}_1, \bar{D}_2, ....\bar{D}_K\}$, $M_0$)

3     **for** $t \leftarrow 1$ **to** $N$ **do**

4         $\bar{h}_{lk} \leftarrow \frac{1}{200} \sum_{x_i \in D_t|_1^{200}} h_{lk}([p_i, x_i])$      `// task-conditioned activation`

5         $\theta_{T_t} \leftarrow \sum_{(l,k) \in \mathcal{S}} \bar{h}_{lk}$

6         $M_i \leftarrow$ `FVGuidedTraining`($D_t, M_{t-1}, \theta_{T_t}, \mathcal{S}$)

7     **end**

8     **return** $M_N$

9  **return**

10 **Function** `GetFunctionVectorSet`($\{D_1, D_2, ....D_K\}$, $M$)**:**

11     $s \leftarrow \text{Array}[:,:,:](0)$

12     **for** $t \leftarrow 1$ **to** $K$ **do**

13         $\bar{h}_{lk} \leftarrow \frac{1}{100} \sum_{x_i \in D_t|_1^{100}} h_{lk}([p_i, x_i])$      `// task-conditioned activation`

14         **for** $l \leftarrow 1$ **to** $LayerNum$ **do**

15             **for** $k \leftarrow 1$ **to** $HeadNum$ **do**

16                 **foreach** $(x, y)$ *in* $D_t |_{100}^{120}$ **do**

17                     $\text{CE}_{lk}([\hat{p}, x]) \leftarrow P_{M^{h_{lk} \to \bar{h}_{lk}}}(y \mid [\hat{p}, x]) - P_M(y \mid [\hat{p}, x])$

18                     $s[t, l, k] \leftarrow s[t, l, k] + \text{CE}_{lk}([\hat{p}, x])$  `// `$\hat{p}$` label-shuffled prompt`

19                 **end**

20             **end**

21         **end**

22     **end**

23     $s_{mean} \leftarrow \frac{1}{K} \sum_{i=1}^{K} s[i,:,:]$               `// average across datasets`

24     $\mathcal{S} \leftarrow$ `GetTop10Indices`($s_{mean}$)

25     **return** $\mathcal{S}$

26 **return**

27 **Function** `FVGuidedTraining`($D, M, \theta_T, \mathcal{S}$)**:**

28     **foreach** $B = (x_i, y_i)$ *in* $GenerateBatches(D)$ **do**

29         $\ell_{LM} \leftarrow \frac{1}{|B|} \sum_{(x,y) \in B} - \log P_M(y \mid x)$      `// language modeling loss`

30         $\ell_{FV} \leftarrow \frac{1}{|B|} \sum_{(x) \in B} \sum_{(l,k) \in \mathcal{S}} d\left(h_{lk}^{M_{t-1}}(x), h_{lk}^M(x)\right)$  `// FV consistency loss`

31         $\ell_{KL} \leftarrow \frac{1}{|B|} \sum_{(x) \in B} KL[P_M(\cdot \mid x) \| P_{M_{t-1}^{h_l \to h_l + \theta_T}}(\cdot \mid x)]$      `// FV-guided`
        `KL-divergence loss`

32         $M.\text{UpdateWeights}(\ell_{LM} + \ell_{FV} + \ell_{KL})$

33     **end**

34     **return** $M$

35 **return**

---

## C   Illustration of causal pathway to forgetting.

To help the understanding of "the causal pathway to forgetting through function vector", we provide the illustrations in Figure 5 and the detailed discussions in Section 5. Refer to the caption of Figure 5 for more information.

## D   Datasets

Three continual instruction tuning benchmarks and severel general evaluation datasets are adopts in this paper. The detailed information is as follows:

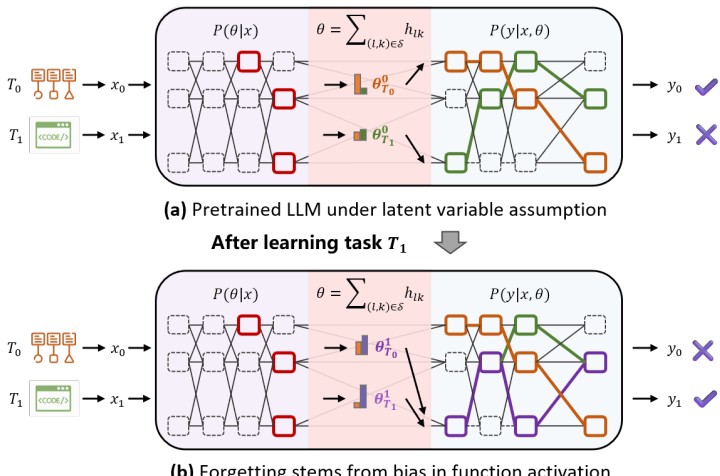

**(a)** Pretrained LLM under latent variable assumption

**After learning task $T_1$**

**(b)** Forgetting stems from bias in function activation

Figure 5: Illustration of causal pathway to forgetting. In (a), the pre-trained model is expressed in a latent variable assumption. It assumes task $T_0$ establishes a predictive pathway (shown in orange) that aligns well with the task (high probability with $\theta^0_{T_0}$). In (b), it shows the model after learning a new task $T_1$ without regularization, which will necessarily update the function attention heads, i.e., $P_M(\theta|x)$, (shown in red blocks), producing new function vectors $\theta^1_{T_0}$ and $\theta^1_{T_1}$ that are biased toward $T_1$. These shifts in function vectors lead to a derailed predictive pathway (shown in purple) with erroneous predictions for task $T_0$; in other words, forgetting of $T_0$ occurs. In summary, the modifications in $P_M(\theta|x)$ rather than $P_M(y|x,\theta)$ are the primary driving force behind forgetting.

Table 3: A summary of dataset statistics in TRACE includes information on the source of the context, average length in terms of word count for English, German, and code datasets, and character count for Chinese.

| Dataset | Source | Category | Avg len | Metric | Language | #data |
|---------|--------|----------|---------|--------|----------|-------|
| ScienceQA | Science | Multi-Choice QA | 210 | ROUGE-L | English | 3,000 |
| FOMC | Finance | Multi-Choice QA | 51 | ROUGE-L | English | 3,000 |
| MeetingBank | Meeting | Summary | 2853 | ROUGE-L | English | 3,000 |
| C-STANCE | Social media | Multi-Choice QA | 127 | ROUGE-L | Chinese | 3,000 |
| Py150 | Github | Code generation | 422 | ROUGE-L | Python | 3,000 |
| NumGLUE-cm | Math | Math reasoning | 32 | ROUGE-L | English | 3,000 |

**TRACE benchmark.** TRACE benchmark is released by Wang et al. (2023c) for the study of forgetting in LLMs, which consists of 8 different complex generation tasks including multi-choice QA, code generation, mathematical reasoning and summary. Without loss of generaliztion, we select 6 out of 8 raw tasks to construct the training sequence as our experiments setup. The statistical information is listed in Table 3.

The training epoch for this benchmark is 5 for C-STANCE, Py150, NumGLUE-cm, 3 for FOMC and ScienceQA, and 7 for MeetingBank. We evaluate them with a self-construct evaluation code based on OpenCompass code framework.

**SuperNI benchmark.** SuperNI benchmark is widely utilized in existing instruction-following works. We select 26 tasks from the original dataset and set the training size to 1000 and training epoch set to 10. The statistical information is listed in Table 5.

**General evaluation sets.** For the general evaluation datasets, we utilize Hellaswag (Zellers et al., 2019), CommonsenseQA (Talmor et al., 2018), OpenbookQA (Mihaylov et al., 2018), Natural Question Kwiatkowski et al. (2019), Lambada Paperno et al. (2016), Alpaca Taori et al. (2023)

Table 4: Basic information of continual learning task sequences used in main text.

| | Sequence | Task type | Num. per task |
|---|---|---|---|
| NI-Seq-C1 | NI195 → NI1343 → NI1310 → NI1292 → NI363 | Classification | 1,000 |
| NI-Seq-C2 | NI231 → NI1343 → NI220 → NI224 → NI273 | Classification | 1,000 |
| NI-Seq-G1 | NI618 → NI1290 → NI589 → NI511 → NI1357 | Generation | 1,000 |
| NI-Seq-G2 | NI1355 → NI141 → NI619 → NI163 → NI002 | Generation | 1,000 |
| NI-Seq-M1 | NI360 → NI363 → NI1290 → NI339 → NI1510 | Classification & Generation | 1,000 |
| NI-Seq-M2 | NI195 → NI611 → NI292 → NI488 → NI024 | Classification & Generation | 1,000 |
| TRACE | Cstance → Fomc → Meet → Py150 → SciQA → Numgluecm | Classification & Generation | 3,000 |

Table 5: A summary of dataset statistics in SuperNI.

| Dataset | Source | Category | Avg len | Metric | Language | #data |
|---|---|---|---|---|---|---|
| NI002 | Quoref | Question Answering | 360 | ROUGE-L | English | 1000 |
| NI1290 | Xsum | Summarization | 363 | ROUGE-L | English | 1000 |
| NI1292 | Yelp review full | Sentiment Analysis | 130 | ROUGE-L | English | 1000 |
| NI141 | Odd man out | Word Semantics | 9 | ROUGE-L | English | 1000 |
| NI273 | Europarl | Text Matching | 15 | ROUGE-L | English | 1000 |
| NI024 | Cosmosqa | Question Answering | 82 | ROUGE-L | English | 1000 |
| NI1310 | Multilingual amazon reviews | Sentiment Analysis | 59 | ROUGE-L | English | 1000 |
| NI163 | Synthetic | Program Execution | 23 | ROUGE-L | English | 1000 |
| NI292 | Storycommonsense | Information Extraction | 48 | ROUGE-L | English | 1000 |
| NI1343 | Amazon us reviews | Sentiment Analysis | 70 | ROUGE-L | English | 1000 |
| NI195 | Sentiment140 | Sentiment Analysis | 14 | ROUGE-L | English | 1000 |
| NI1355 | Sentence compression | Summarization | 25 | ROUGE-L | English | 999 |
| NI589 | Amazon fine food reviews | Summarization | 84 | ROUGE-L | English | 1000 |
| NI1357 | Xlsum | Summarization | 454 | ROUGE-L | English | 1000 |
| NI360 | Numersense | Fill in The Blank | 26 | ROUGE-L | English | 1000 |
| NI339 | Record | Question Answering | 185 | ROUGE-L | English | 1000 |
| NI220 | Rocstories | Title Generation | 60 | ROUGE-L | English | 1000 |
| NI224 | Scruples | Ethics Classification | 338 | ROUGE-L | English | 1000 |
| NI611 | Mutual | Dialogue Generation | 162 | ROUGE-L | English | 1000 |
| NI1510 | Evalution | Information Extraction | 7 | ROUGE-L | English | 1000 |
| NI231 | Iirc | Question Answering | 229 | ROUGE-L | English | 1000 |
| NI488 | Synthetic | Program Execution | 16 | ROUGE-L | English | 1000 |
| NI618 | Multilingual amazon reviews | Summarization | 47 | ROUGE-L | English | 1000 |
| NI363 | Sst2 | Sentiment Analysis | 19 | ROUGE-L | English | 1000 |
| NI619 | Ohsumed | Title Generation | 161 | ROUGE-L | English | 1000 |
| NI511 | Reddit tifu dataset | Summarization | 400 | ROUGE-L | English | 1000 |

Table 6: Input template for calculating instruction probability and training for different tasks.

| Task | Prompts |
|------|---------|
| ScienceQA | "Input": "Choose an answer for the following question and give your reasons. Question: $[x]$ Answer:", "Output": "$[y]$" |
| FOMC | "Input": "What is the monetary policy stance for the following text? A. dovish, B. hawkish, C. neutral. Choose one from A, B and C. Text: $[x]$ Stance:", "Output": "$[y]$" |
| C-STANCE | (Translate Chinese to English) "Input": "Determine the attitude of the following text towards the specified object. Select one: A. Support, B. Oppose, C. Neutral. Output A, B or C. Text: $[x_1]$ Object: $[x_2]$ Attitude:", "Output": "$[y]$" |
| MeetingBank | "Input": "Write a summary of the following meeting transcripts. Meeting transcripts: $[x]$ Summary:", "Output": "$[y]$" |
| Py150 | "Input": "¡s¿ [x]", "Output": "$[y]$" |
| NumGLUE-cm | "Input": "Solve the following math problem. Question: $[x]$ Answer:", "Output": "$[y]$" |
| NI-xxx | "Input": "Definition: In this task, you're given $[Description]$. Now complete the following examples Input: $[x]$ Output:", "Output": "$[y]$" |

and Bbh-Object Count Srivastava et al. (2022). All the datasets is downloaded from `https://github.com/open-compass/opencompass` and truncate to 190 samples for efficiency.

**Input template** In this paper, the specific instruction template used for each dataset is given below, as show in Table 6.

## E  IMPLEMENTATION

We adopt Llama2-7b-chat, Llama2-13B-chat (Touvron et al., 2023b), Llama3-8B-chat (Dubey et al., 2024), Mistral-7B-instruct-v2.0 (Jiang et al., 2023) as the base models, with their effectiveness in both understanding world knowledge and following instructions. Without specific notification, the model is fine-tuned with LORA approach Hu et al. (2021), where the rank dimension set to 8 and the target module is query and value weight matrices. For IncLora, OLora, and InsCL methods, a new adapter is initialized at the beginning of learning new task while keep the previous Lora adapters fixed. For Ewc, only one big adapter is initialized during the sequential learning, where rank is set to 48 for TRACE, and 40 for NI benchmarks.

The maximum input sequence length is set to 512 and the maximum output sequence length is set to 128. We train the model with the decoder only task calculating gradient only on the output tokens. We use an Adam optimizer with a weight decay of 1e-4 and the learning rate set to 1e-4 for TRACE and FUNC, 1e-3 for LONG (following Wang et al. (2023c)). The batch size is set to 8 and accumulate gradient step is set to 2 for each GPU while we run on 4 A100 GPUs with Deepspeed. The training size and epochs can be found in the introduction of datasets.

**Implementation Detail of Optimization Objective**

In order to enhance the reproducibility of the paper, we provide the detailed calculation formula for the loss function $\ell_{FV}$ and $\ell_{KL}$ when training task $T_j$:

$$\ell_{FV} = \sum_{(l,k)\in\mathcal{S}} \|h_{lk}^{M_{j-1}}(x) - h_{lk}^{M}(x)\|_2^2 \tag{8}$$

$$\ell_{KL} = \sum_{i=1}^{V} P_M(\mathcal{Y}_i \mid x)[\log P_M(\mathcal{Y}_i \mid x) - \log P_{M_{j-1}^{h_l \to h_l + \theta_{T_j}}}(\mathcal{Y}_i \mid x)] \tag{9}$$

Here, $P_M(\mathcal{Y}_i \mid x)$ denotes the output probability of token $\mathcal{Y}_i$ and $V = |\mathcal{Y}|$ is the vocabulary size. As for the hyper-parameters $\alpha_1$ and $\alpha_2$, we perform a grid search on [2, 1, 0.5, 0.25, 0.08, 0.02] and set $\alpha_1 = 1$ and $\alpha_2 = 0.08$ as the final choice.

Table 7: Final performance on 3 SuperNI benchmarks on 4 language models. Hella., Com., Alpa., and Ob. denote evaluation score on Hellswag, CommonsenseQA, Alpaca, Object Count datasets, respectively. The **Del.** value in red bold style is compared to performance of their initial model $M_0$. Higher **Forget** or lower **Del.** represent more forgetting. ***Main conclusion:** Forgetting consistently occurs in both general and newly learned tasks, showing considerable variations depending on the types of tasks, stages of training, and the specific language models involved.*

| | Zero-Shot Performance in General Task | | | | | In-Context Performance in General Task | | | | | Performance in Trained Task | | |
|---|---|---|---|---|---|---|---|---|---|---|---|---|---|
| | Hella. | Com. | Alpa. | Ob. | Avg./**Del.** | Hella. | Com. | Alpa. | Ob. | Avg./**Del.** | AP | FP | **Forget** |
| **Llama2-7b-chat** | | | | | | | | | | | | | |
| $M_0$ | 57.89 | 57.37 | 26.50 | 27.12 | 42.22 | 58.95 | 57.89 | 35.17 | 34.21 | 46.56 | / | / | / |
| NI-Seq-C1 | 47.37 | 40.00 | 32.00 | 31.61 | 37.75 | 24.21 | 27.89 | 28.89 | 26.84 | 26.96 | 86.10 | 83.80 | 2.30 |
| NI-Seq-C2 | 65.26 | 55.79 | 30.99 | 23.47 | 43.88 | 67.37 | 54.21 | 32.66 | 27.89 | 45.53 | 91.80 | 88.10 | 3.70 |
| NI-Seq-G1 | 48.95 | 39.47 | 27.36 | 39.72 | 38.88 | 37.89 | 42.63 | 28.84 | 38.95 | 37.08 | 24.97 | 19.36 | 5.61 |
| NI-Seq-G2 | 48.42 | 32.11 | 26.30 | 31.05 | 34.47 | 17.89 | 27.89 | 15.37 | 38.95 | 25.03 | 49.42 | 43.37 | 6.06 |
| NI-Seq-M1 | 52.11 | 42.63 | 31.09 | 29.51 | 38.84 | 45.79 | 31.05 | 24.58 | 33.16 | 33.65 | 59.02 | 54.33 | 4.70 |
| NI-Seq-M2 | 65.26 | 43.16 | 30.08 | 37.09 | 43.90 | 44.21 | 27.89 | 27.64 | 37.37 | 34.28 | 73.19 | 55.79 | 17.40 |
| **Llama3-8b-chat** | | | | | | | | | | | | | |
| $M_0$ | 81.58 | 58.42 | 22.64 | 40.04 | 50.67 | 85.26 | 63.16 | 27.42 | 49.47 | 56.33 | / | / | / |
| NI-Seq-C1 | 79.47 | 46.84 | 23.27 | 32.32 | 45.48 | 79.47 | 40.00 | 25.62 | 45.79 | 47.72 | 83.40 | 82.10 | 1.30 |
| NI-Seq-C2 | 80.53 | 58.42 | 23.62 | 42.19 | 50.54 | 84.21 | 50.00 | 24.70 | 44.25 | 50.78 | 91.00 | 89.90 | 1.10 |
| NI-Seq-G1 | 72.63 | 35.79 | 22.05 | 29.39 | 39.97 | 67.89 | 31.05 | 19.77 | 41.58 | 40.07 | 28.29 | 21.10 | 7.20 |
| NI-Seq-G2 | 63.68 | 41.58 | 20.9 | 15.37 | 35.38 | 66.84 | 44.74 | 15.35 | 23.58 | 37.63 | 57.77 | 55.66 | 2.11 |
| NI-Seq-M1 | 78.42 | 40.00 | 21.93 | 21.58 | 40.48 | 76.84 | 40.00 | 21.32 | 35.91 | 43.52 | 60.74 | 52.63 | 8.11 |
| NI-Seq-M2 | 80.53 | 58.42 | 20.95 | 42.28 | 50.55 | 74.21 | 51.05 | 23.93 | 19.79 | 42.26 | 74.49 | 52.34 | 22.16 |
| **Mistral-7b-instruct** | | | | | | | | | | | | | |
| $M_0$ | 73.68 | 60.00 | 24.74 | 5.02 | 40.86 | 79.47 | 66.32 | 32.36 | 37.89 | 54.01 | / | / | / |
| NI-Seq-C1 | 63.16 | 50.00 | 32.04 | 15.30 | 40.13 | 66.84 | 51.05 | 36.80 | 37.89 | 48.15 | 84.70 | 85.40 | -0.70 |
| NI-Seq-C2 | 75.79 | 60.00 | 32.07 | 36.63 | 51.12 | 73.68 | 58.95 | 35.76 | 37.37 | 51.44 | 91.50 | 90.30 | 1.20 |
| NI-Seq-G1 | 57.37 | 45.26 | 26.30 | 13.81 | 35.69 | 57.89 | 35.79 | 32.04 | 39.47 | 41.30 | 27.63 | 19.78 | 7.85 |
| NI-Seq-G2 | 33.68 | 42.63 | 29.68 | 52.11 | 39.53 | 41.58 | 36.32 | 20.46 | 30.53 | 32.22 | 51.05 | 43.86 | 7.19 |
| NI-Seq-M1 | 65.26 | 47.89 | 33.02 | 12.35 | 39.63 | 63.68 | 38.42 | 34.79 | 45.79 | 45.67 | 61.96 | 57.01 | 4.96 |
| NI-Seq-M2 | 57.37 | 48.42 | 31.67 | 35.58 | 43.26 | 67.37 | 47.89 | 34.72 | 46.53 | 49.13 | 72.22 | 65.95 | 6.27 |
| **Llama2-13b-chat** | | | | | | | | | | | | | |
| $M_0$ | 69.47 | 51.05 | 28.99 | 15.09 | 41.15 | 75.26 | 57.89 | 35.46 | 43.16 | 52.94 | / | / | / |
| NI-Seq-C1 | 65.79 | 52.63 | 34.18 | 21.51 | 43.53 | 66.32 | 48.42 | 38.48 | 38.95 | 48.04 | 83.20 | 82.27 | 0.93 |
| NI-Seq-G1 | 63.16 | 38.95 | 28.12 | 13.84 | 36.02 | 65.79 | 32.11 | 30.92 | 34.21 | 40.76 | 25.64 | 18.17 | 7.47 |
| NI-Seq-M1 | 71.58 | 49.47 | 34.10 | 28.09 | 45.81 | 70.53 | 48.42 | 36.51 | 37.37 | 48.21 | 60.10 | 56.34 | 3.76 |

For the hyperparameters of existing continual learning methods, we refer to the well-searched value reported in previous paper. Specifically, for Ewc the scaling factor on regularization term is set to 4,000, for O-lora the number is 0.5. The memory size of InsCL is set to 30 for NI benchmark and 50 for TRACE.

**Implementation Detail of Function Vector Framework** When extracting the function vector from in-context samples, we use 10-shot input prompt randomly selected from held-out training dataset. The task-conditioned activations are average on samples filtered with correct 10-shot answer from the validation set with 200 samples. As for the set $\mathcal{S}$ of the casual attention heads, we follow the position in Todd et al. (2023) for Llama2-7b-chat and Llama2-13b-chat, and validate its efficiency on our own datasets. Specifically, for Llama2-7b-chat, the set $\mathcal{S}$ is $[(14, 1), (11, 2), (9, 25), (12, 15), (12, 28), (13, 7), (11, 18), (12, 18), (16, 10), (14, 16)]$. For Llama2-13b-chat the set $\mathcal{S}$ is $[(13, 13), (12, 17), (15, 38), (14, 34), (19, 2), (19, 36), (13, 4), (18, 11), (10, 15), (13, 23)]$. As for Llama3-8b-chat and Mistral-7b-instruct, we run the casual analysis experiments on 15 datasets and calculate the average CE to get the final casual attention heads set. For Llama3-8b-chat, the set $\mathcal{S}$ is $[(27, 28), (13, 27), (15, 28), (17, 8), (21, 2), (10, 12), (15, 16), (15, 2), (15, 1), (31, 24)]$. For Mistral-7b-instruct, the set $\mathcal{S}$ is $(14, 31), (26, 29), (12, 4), (12, 7), (30, 4), (30, 9), (22, 30), (14, 19), (11, 10), (18, 1)]$.

## F    EXTENDED EXPERIMENTAL RESULTS

**Results on more sequences.** In addition to the results of the three sequences presented in Sec. 3, we conducted the similar experiments on more sequences and found forgetting patterns similar to the experimental results described in the main text. The detailed experimental results on totally 6 sequences are shown in Table 7.

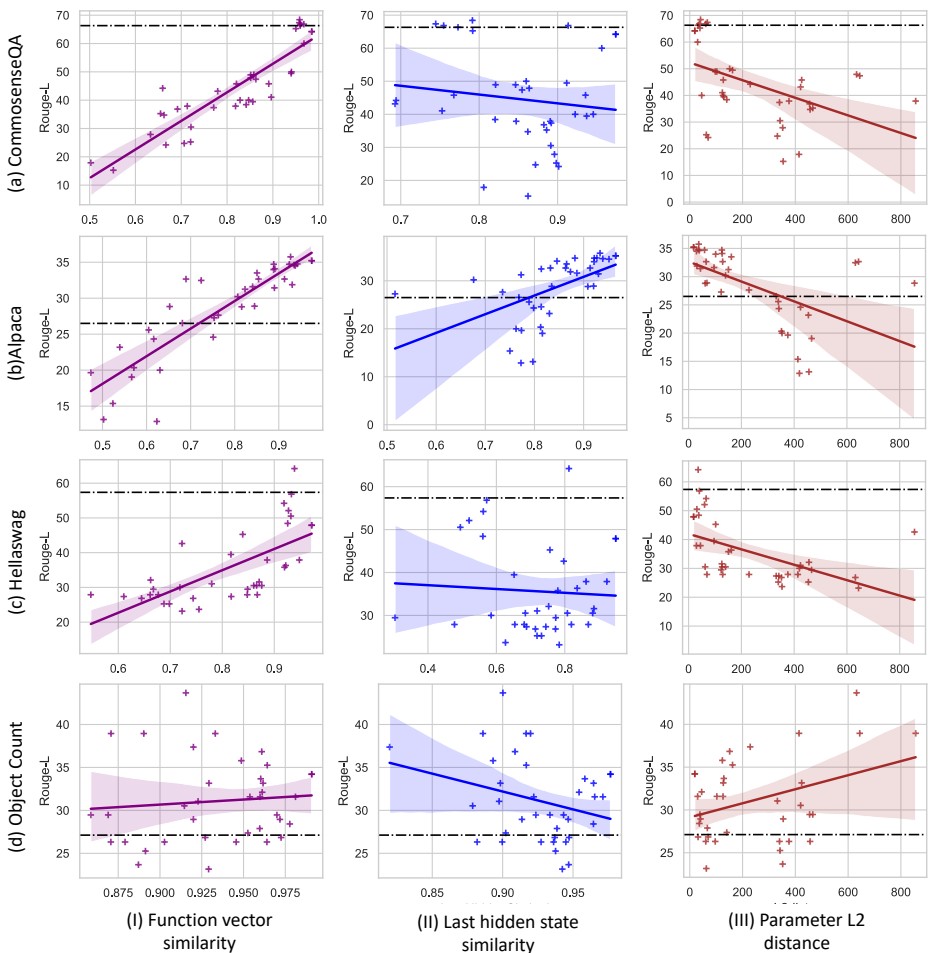

Figure 6: The correlation plot on model performance and different similarity metrics. The y-axis shows Rouge-L metric on test data, while the x-axis represents the degree of similarity between the current model state and its initial condition. The calculation of each similarity metrics is (1) FV similarity: $\text{Cosine}(\theta_{T^e}^0, \theta_{T^e}^j)$. (2) Last hidden state similarity: $\text{Cosine}(\sum h_{-1,-1}^0(x), \sum h_{-1,-1}^j(x))$. (3) Parameter L2 distance: $\|W^j - W^0\|^2$. The dotted line in each figure denotes the performance for original model. ***Main conclusion:*** *There is a significant correlation between performance and FV similarity (sub-figures in the first column), while the other two metrics—last hidden state similarity and L2 distance—do not show such strong correlation.*

**Correlation plot on model performance and different similarity measurements.** In this section, we provide scatter plots to illustrate the correlation between model performance and function vector (FV) similarity, alongside two other metrics: the similarity of the last layer hidden states and the L2 distance of parameters. The calculations for each similarity measure are as follows:

FV similarity is calculated using $\text{Cosine}(\theta_{T^e}^0, \theta_{T^e}^j)$, where $\theta_{T^e}^j$ represents the FV of the evaluation task $T^e$ after fine-tuning the $j$-th task. Last layer hidden states similarity is derived from $\text{Cosine}\left(\sum_{x \in E} h_{-1,-1}^0(x), \sum_{x \in E} h_{-1,-1}^j(x)\right)$, where $E$ is the test dataset and $h_{-1,-1}^j$ denotes the model's output representation at the last token position in the last layer after fine-tuning the $j$-th task. Parameter L2 distance is defined as $\|W^j - W^0\|^2$, with $W^j$ being the model weight post fine-tuning the $j$-th task. Here the reported model performance is te 5-shot results on the evaluation dataset.

For each evaluation task, we collected 40 data points from various models across different task sequences and stages and created correlation diagrams. The results, detailed in Figure 6, demonstrate a notable correlation between model performance and FV similarity. This is particularly significant for tasks like Hellaswag, CommonsenseQA, and Alpaca, where a decrease in similarity corresponds

Table 8: Performance of baselines and their improved version with Function Vector Guided (**FVG**) training or Model Averaging (**MA**). ***Main conclusion:*** *MA performance better on fine-tuned datasets (**FP**) compared to FVG, but struggles in the general/in-context datasets setting (**GP/IP**).*

| | Method | NI-Seq-G1 | | | NI-Seq-C1 | | | NI-Seq-M1 | | |
|---|---|---|---|---|---|---|---|---|---|---|
| | | GP ↑ | IP ↑ | FP ↑ | GP ↑ | IP ↑ | FP ↑ | GP ↑ | IP ↑ | FP ↑ |
| Llama2-7b-chat | $M_0$ | 49.85 | 54.43 | | 49.85 | 54.43 | | 49.85 | 54.43 | |
| | LoraInc | 47.16 | 30.94 | 19.35 | 45.83 | 27.71 | 83.80 | 47.55 | 37.23 | 54.33 |
| | +MA | +0.87 | +10.35 | **+1.46** | +2.81 | +16.55 | -0.17 | **+3.90** | +9.95 | +2.22 |
| | +FVG | **+3.10** | **+18.97** | +0.84 | **+3.98** | **+25.53** | **+1.70** | +2.65 | **+15.78** | **+3.52** |
| | Ewc | 33.48 | 26.87 | 17.72 | 46.08 | 38.76 | 85.00 | 44.47 | 41.69 | 55.85 |
| | +MA | +6.58 | +11.27 | **+2.59** | +1.57 | +7.64 | **+0.40** | +5.54 | +7.71 | **+0.92** |
| | +FVG | **+15.73** | **+27.18** | +0.85 | **+3.11** | **+15.96** | +0.37 | **+6.18** | **+13.99** | +0.01 |
| Llama3-8b-c. | $M_0$ | 56.61 | 60.61 | | 56.61 | 60.61 | | 56.61 | 60.61 | |
| | LoraInc | 45.51 | 39.85 | 21.10 | 51.89 | 54.63 | 82.10 | 48.00 | 47.82 | 52.63 |
| | +MA | +4.39 | +8.01 | +2.07 | +1.99 | +2.42 | **+2.00** | +3.67 | **+5.82** | +4.70 |
| | +FVG | **+7.79** | **+15.31** | **+3.10** | **+3.99** | **+5.19** | +0.30 | **+4.88** | +4.75 | **+5.78** |

to an increase in model forgetting. However, for scenarios where no forgetting occurs, such as in Object Count, there is no apparent correlation between FV similarity and performance. These observations inspire further investigations into the mechanisms of task transfer in LLMs.

Contrarily, the other two metrics—last hidden state similarity and L2 distance—do not show such strong correlation, indicating their limited effectiveness in reflecting model forgetting.

**Relationship between forgetting and changes in FV similarity between tasks.** In Figure 7, we present the similarity between the FVs of training and evaluation tasks, alongside the corresponding forgetting after training. This similarity between FVs is defined as $\text{Cosine}(\theta_{T^e}^{j-1}, \theta_{T_j}^{j-1})$, where $\theta_{T^e}^{j-1}, \theta_{T_j}^{j-1}$ are the function vectors extracted from model $M_{j-1}$ on evaluation task $T^e$ and the current training task $T_j$, respectively. We observe a non-trivial phenomenon: the lower the similarity between the FVs of two tasks, the greater the extent of forgetting on the evaluation task after training. For instance, in the first column of Figure 7, the most severe forgetting occurs at task $T_2$ while function vectors exhibit low similarity. This contrasts with prior findings (Ramasesh et al., 2020; Lee et al., 2021), where higher task feature similarity was linked to greater forgetting. We hypothesize that this phenomenon stems from the substantial capacity and universality of LLMs, enabling them to construct new functions based on old ones without overwriting as continual instruction tuning proceeds. The lower similarity between the FVs of the training and evaluation tasks indicates more diverse functions introduced, which exacerbates the challenge of locating the groundtruth function corresponding to the evaluation task and thus leads to forgetting. We defer proof of this hypothesis to Section 5.

**Relationship between forgetting and hidden states similarity.** In Figure 7, we present the similarity between the FVs of training and evaluation tasks, alongside the corresponding forgetting after training. To further verify that simple feature similarity is insufficient to represent the forgetting phenomenon, we also include a heatmap of last layer hidden states similarity between training and testing tasks. This representational similarity was obtained through $Cosine(\sum_{x \in T} h_{-1}^{-1}(x), \sum_{x \in E} h_{-1}^{-1}(x))$, where $T, E$ represent the training and testing tasks, respectively, and $h_{-1}^{-1}$ represents the model's output representation at the last input token position of the last layer. By comparing the first and third rows in Figure 7, we were unable to identify any significant correlation between them, indicating that relying solely on simple model representations to study the forgetting phenomenon is not advisable.

**Comparison between model averaging.** To further demonstrate the advanced nature of our algorithm, we compared FVG with Model Averaging (Lin et al., 2024). Model averaging is a technique often used to improve the robustness and performance of models. It involves taking the average of multiple model parameters across different training runs or stages and has been proven for its effectiveness in mitigating forgetting.

We evaluate Model Averaging on three benchmarks with combination of IncLora and EWC methods on Llama2-7b-chat and Llama3-8b-instruct models. Specifically, we perform Model Averaging on pre-trained model and final model with the averaging ratio set to 0.2. The results are shown in

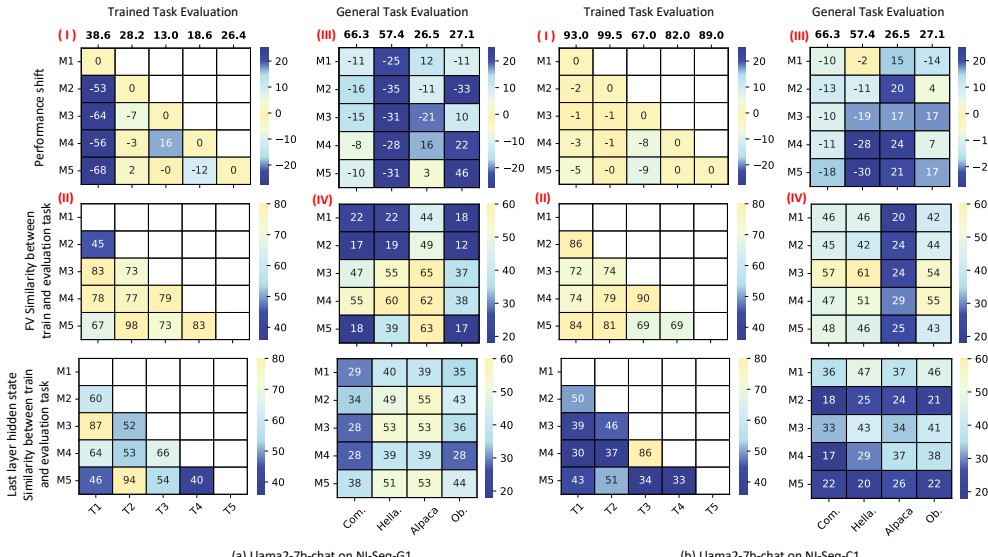

Figure 7: Heatmaps of performance shift (Top) and function vector similarity (Bottom) between training and test tasks before tuning. For performance shift, the value at position $(j', j)$ represents the percentage change of task $j$ at moment $M_{j'}$ relative to the baseline metrics. For FV similarity, the value at position $(j', j)$ corresponds to $\mathrm{Cosine}(\theta_{T^e}^{j'}, \theta_{T_j}^{j'})$. **Main conclusion:** *Lower FV similarity (bluer value in the second row table) between tasks correlates with increased forgetting (bluer value in the first row table) after training; however, similarity in hidden states does not demonstrate this correlation.*

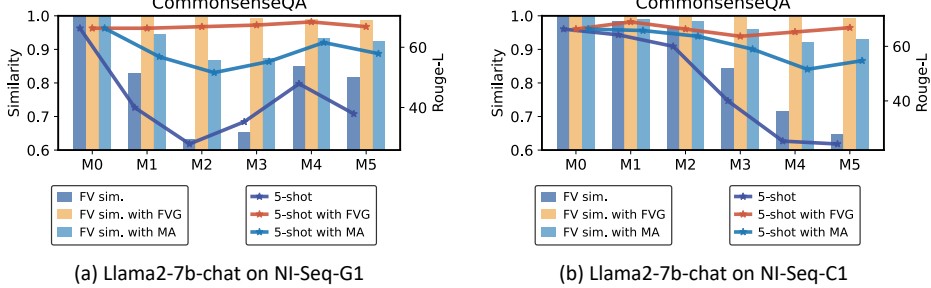

Figure 8: The shifts in function vector with 5-shot performance with function vector guided training (FVG) and model averaging (MA). **Main conclusion:** *FVG and MA prevents the shift in FV (yellow and light blue bar) and thus mitigating forgetting (orange and light blue line).*

Table 8. It shows a better performance on fine-tuned datasets (**FP**) compared to function vector guided training, but struggles in the general/in-context datasets setting (**GP/IP**).

While Model Averaging contributes to avoiding forgetting, it is interesting to see the explanation in the perspective of function vector. We provide the shift in function vector before and after model averaging with corresponding performance in Fig. 8. Cross-method comparative analysis shows that methods capable of maintaining the stability of FV changes tend to yield better results. Specifically, model averaging, when compared to its predecessor Inclora, mitigates shifts and enhances performance. Furthermore, an examination across various training stages indicates a positive correlation between performance and the extent of FV shifts including Model Averaging.

**Effectiveness of Function Vector**    To assess the effectiveness of the extracted $\theta_T$, referred to as the Function Vector (FV) in this study, we conduct a series of intervention experiments across multiple datasets (see Fig. 9) on the initial model Llama2-7b-chat. These experiments consisted of either inserting or removing an FV at the hidden states of a specific layer at the the last token position, to

examine the influence on the model output. More precisely, in the transformer's forward residual stream, the instruction vector $\theta_T$ modifies the hidden states at a select layer $l$ as $h_l = h_l + \theta_T$.

We reported the intervention findings on four distinct datasets: 1) CommensenseQA (different from the evaluation set mentioned above in input instruction), multiple-choice questions on common sense reasoning; 2) Antonym, a task aimed at generating antonyms; 3) AGNews, a text classification task with the article's category as the label; and 4) Last-Spanish, a task that output the Spanish translation of the list's final item. The results highlighted that the FV directly affects the model's output behavior for specific tasks. In tasks such as Antonym, Last-Spanish, and CommonsenseQA, introducing FV significantly improved the zero-shot performance from a low level. Conversely, in the cases of AGNews and CommonsenseQA, removing the FV resulted in a deterioration of the model's ability to produce the correct output. In contrast, interventions with random vectors had a negligible effect on the model.

**Alternation of casual attention head during training.** We carried out causality analysis experiments to identify the latest causal attention head $\mathcal{S}$ in the

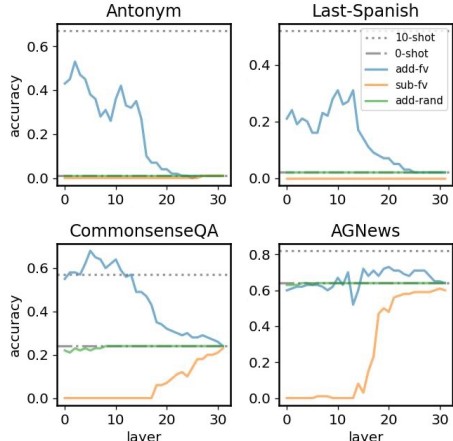

Figure 9: Intervention results on four datasets via function vector. ***Main conclusion:*** *Function is effective in regulating the final outputs.*

model, which was fine-tuned on NI-Seq-G1. Specifically, these causality analysis experiments were performed across six datasets, and the average Cross-Entropy (CE) was calculated to determine the final set of causal attention heads. The findings are presented in Fig. 10. We observed a gradual yet slight shift in the set $\mathcal{S}$. This indicates that changes in the model's function vector occur not only in values but also in positions, though such changes are slow and do not significantly alter the importance of the original positions. Therefore, the function vectors mentioned in this paper are extracted from uniform positions.

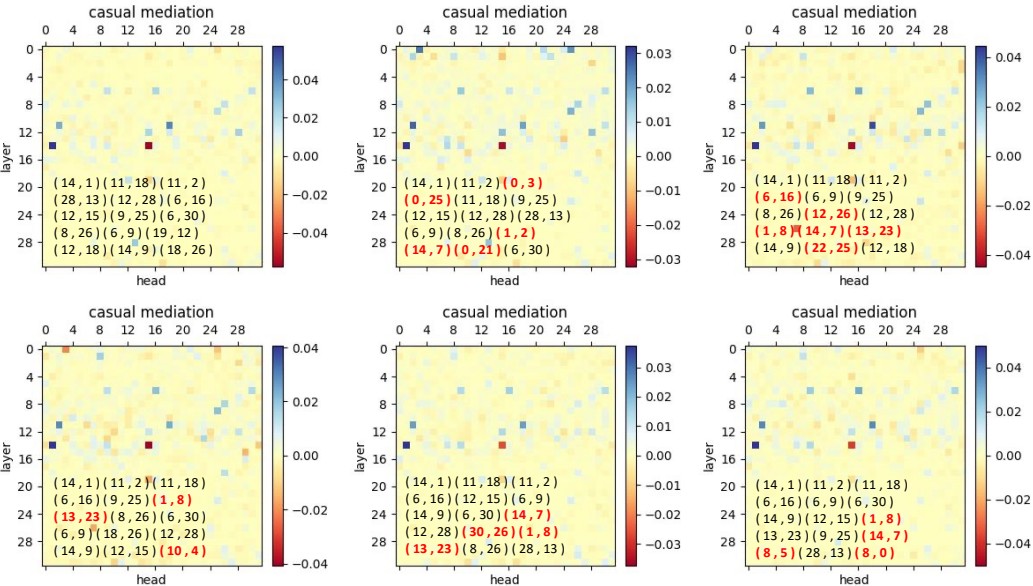

Figure 10: Alternation in the casual attention head during Llama2-7b-chat training on NI-Seq-G1. The positions of top-10 heads are listed in each heatmap, while the newly introduced heads are marked as red. **Main results:** *The position of function vector shifts at a quite slow speed during training*

