# OpenReview forum: "Unlocking the Power of Function Vectors for Characterizing and Mitigating Catastrophic Forgetting in Continual Instruction Tuning"
_ICLR.cc/2025/Conference — ICLR 2025 Oral_

### Official Review · Reviewer_Qiar · 2024-10-25

**Soundness:** 4
**Presentation:** 4
**Contribution:** 4
**Rating:** 10
**Confidence:** 4

**Summary:**

This paper studies the catastrophic forgetting (CF) problem in LLMs with the tool of Function Vector (FV). The research starts with reproducing the CF problem on a series of four tasks across 4 publicly available LLMs. In addition, the researchers study the differences in FVs between continue-trained and untrained LLMs on each task. They show that there is a strong correlation between the differences in FVs and the CF issue. Their theoretical analysis posts a hypothesis on the bias of the FV θT elicited by the input x, (i.e., p(θT|x)). The authors further conducted several experiments to well support their hypothesis. Finally, they designed a new loss for alleviating the CF problem in continuing learning, where they encourage LLMs to minimize the differences between model prediction and the FV-intervented prediction while keeping the hidden representations of X the same as un-trained ones.

**Strengths:**

1. The manuscript provides a thorough study of the catastrophic forgetting problem.
2. The most insightful contribution of this work is posing a new hypothesis on this problem.
3. Its technical contribution is significant as the authors first introduce the Function Vector as a tool in this community.
4. The paper is well-written and well-organized.

**Weaknesses:**

I cannot see any significant weakness. One nitpick would be expanding the background of Function Vector in Sec. 2.2, especially providing the dimension or shape of each notation. It will help the audiences understand the procedure for computing FVs more easily.

**Questions:**

Here are some minor/typo suggestions:
1. Line 107, the references to Peng et al. and Chung et al. should be \citep instead of \cite.
2. Line 143, missing a blank between "using" and "SuperNI".
3. Line 162, the reference to Zhao et al. should be \citep instead of \cite.
4. Line 200, missing a blank between "." and "Unless".
5. Line 200, the reference to Hu et al. should be \citep instead of \cite.
6. Line 201, "learning rate of 1e-4" should be "learning rate of $1e^{-4}$".
7. Line 143/157/197, appending a "." after each boldface phrase will be more consistent with your writing style in Line 240/249/257 and so on.
8. Line 357,  "... trigger its corresponding latent variable θT (PM(θT|x))." could be rewritten as "... trigger its corresponding Function Vector θT (i.e., PM(θT|x))." for more clarification.
9. It would be great to provide a one-sentence summary at the end of each paragraph in Section 7 about the connection between the related works and this study.
10. The Conclusion section seems to be more consistent with the full manuscript if it is described in the present tense.

---

> ### Author Response · Authors · 2024-11-23
> **Response to Reviewer Qiar**
>
> Thank you sincerely for your thoughtful and positive feedback on our work. We are particularly grateful for your recognition of the various aspects of our research. We hope that our responses and revisions adequately address the concerns you've raised. Please feel free to let us know if you have any additional concerns or questions.
>
> **W1:** Expand the background of Function Vector in Sec. 2.2.
>
> > Following your suggestion, we first added the description of the notation shape in Section 2 of the main text and further clarified the notation **throughout the document**. We have revised our paper to include these details.
>
>
> **Q1:** There are some suggestions/typos.
>
> > Thank you for your patience and for pointing out the above-mentioned typos in our paper. We truly appreciate your valuable feedback and will make sure to **revise our paper thoroughly**.
> > We **acknowledge your suggestion** for improving the writing of the paper:
> > - We have added a one-sentence summary at the end of each paragraph in Section 7 to better connect the related works with our study.
> > - We have revised the Conclusion section to use the present tense for consistency with the rest of the manuscript.

---

> > ### Comment · Reviewer_Qiar · 2024-11-23
> > **Comment to Author Response**
> >
> > My concerns have been addressed.

---

> ### Author Response · Authors · 2024-11-25
> **Official Comment by Authors**
>
> We are thrilled to note that your key concerns have been effectively addressed. We sincerely appreciate your dedicated time and effort in examining our paper and offering invaluable and positive feedback.

---

### Official Review · Reviewer_ZyxB · 2024-10-28

**Soundness:** 3
**Presentation:** 2
**Contribution:** 3
**Rating:** 10
**Confidence:** 3

**Summary:**

This paper empirically studies catastrophic forgetting (CF) from the perspective of function vectors (FV) and makes the following core contributions:
1. It shows that there is a correlation between the change in function vector for a task and degradation in performance on the task.
2. It argues that CF is due to the inability to associate the input with the latent task vector rather than a degradation in the ability to solve a tasks conditioned on the input and correct latent task vector. It supports this with mechanistic interventions to the task vector.
3. It presents a method to regularize FVs during training to mitigate forgetting.

**Strengths:**

**Originality**: This was an interesting approach/hypothesis for studying forgetting that I quite enjoyed.
**Quality**: The authors perform a thorough series of experiments investigating their claims and evidence ranges from empirical measurements, mechanistic interventions, and training interventions.
**Clarity**: The work is well contextualized in related work.
**Significance**: This work is a really interesting way to connect mechanistic interpretability to studying and mitigating forgetting demonstrating practical ways in which tools from mechanistic interpretability can be used to understand diverse phenomena.

**Weaknesses:**

1. **Clarity**: The paper dense and the writing is terse and sometimes hard to follow. The figures have a lot of information on them, are very small and it is quite difficult to extract takeaways from them. Overall the paper is often hard to follow and it is hard to trace the evidence for the claims. A lot of points through the paper seem like observations and the connection to one of the core scientific claims isn't always clear. I think this paper would benefit greatly from better organization. Probably most important is to explain the figures/tables better *in the captions* and actually point out what the takeaways are/what we should learn from them/what points it is substantiating. I understand this is challenging because of page constraints and the amount of information presented in the paper: my recommendation would be to move anything non-essential to the core scientific claims to the appendix. Also, the authors could present the results for 1 model and add the rest of the models to the appendix given that the models are all of the same scale.
2. **FV and forgetting**: One of the key claims in this paper is change in FV is correlated with forgetting, i.e. this is a useful measure that "explains" the phenomena of forgetting. The evidence did not convince me of this: in Figure 2, there are points where FV similarity is high, 5 shot performance is high but 0-shot performance is low. Object Count especially is unconvincing (0shot and 5shot switch and don't seem to be correlated with FV).
3. **Mechanistic interventions**: While the mechanistic interventions in Section 5 are quite interesting, the paper does not consider any alternated explanations for this behavior (see questions). It is also not clear to me why FV is a measure of latent task identification.
4. **Limitation of training method**: If I understand correctly, this training method regularizes the FV of particular tasks and thus prevents forgetting of the particular tasks being regularized. It doesn't affect forgetting as a whole? While this is not a weakness, I think this should be clarified and the method should be explained as such.

**Questions:**

1. Related to weakness 2: can you more concretely establish the FV similarity is correlated with forgetting, perhaps with a correlation plot? I want to better understand how much signal FV provides on forgetting as a whole.
2. Related to weakness 2: can you show how FV measures forgetting compared to other metrics one might consider. For example, after training on a task, does FV similarity to the previous task better predict forgetting of the previous task than L2 distance to the previous model?
3. Related to weakness 3: Why is FV a measure of latent task identification? What exactly is the argument for this?
4. Related to weakness 3: an alternate hypothesis for the results in section 5 is that after finetuning, only certain layers of the model have changed significantly while a lot of the other layers have remained unchanged. So when you make the activation intervention and search over the layers to find the optimal one, you have just undone the effect of finetuning in the activations by returning it to the behavior of an earlier model upto that later and the rest of the model isn't that different after finetuning so activations pass through and your recover the behavior of the earlier model. How would you refute this alternate hypothesis?
5. Related to weakness 4: did I understand this correctly? If so, can you make this clear/provide some evidence that if you train with the activations for task 1 regularized, it does not mitigate forgetting on task 2 (another unrelated task). If this is true, this is an important limitation that should be made clear (I want to emphasize that it isn't a limitation of the paper but rather would make the paper stronger be explaining how this method should be used).
6. A claim for the training method is that it does not effect plasticity. I would like to understand the limitations of this: can we finetune on a very large dataset of a new task with this regularization and still maintain plasticity? When does the regularization start hurting? What are the limits. One way you could do this is train on a FT dataset with millions of samples (maybe OpenMathInstruct-2) to see if this regularization limits learnability in extreme cases.

---

> ### Author Response · Authors · 2024-11-23
> **Response to Reviewer ZyxB**
>
> Thank you for your thoughtful comments and for taking the time to review our paper. Below, we have included detailed responses to your feedback, along with revisions to the manuscript highlighted in blue for your convenience. We hope our replies and revisions sufficiently address your concerns and enhance clarity. Please do not hesitate to reach out if you have any further questions or feedback.
>
> **W1:** Explain the figures/tables better *in the captions* and actually point out what the takeaways are/what we should learn from them/what points it is substantiating.
>
> > Thank you for your thorough review and valuable suggestions. We now **revised the caption** of the figures/tables in the paper with **an explanation of the main conclusion** pointing out the main takeaways with the **substantive supporting data**.
> > - For example, we added "Main conclusion: (1) Learning generation tasks (a/c) vs. classification tasks (b/d) lead to more forgetting.; (2) Forgetting may reduce naturally (a-(II)/d-(II)); (3) Forgetting is model-dependent (a/b vs. c/d)." to the caption of Figure 2 for better clarity.
> >
> > Furthermore, we have made several other revisions to enhance the clarity and readability of the paper:
> >
> > - We have now included a detailed description of the **training algorithm** used in our study in Appendix B.
> > - We have added illustrations in Figure 5 and detailed discussions in Section 5 Line 336-352 of **the causal pathway contributing to forgetting**, along with the motivations behind our Function Vector Guided (FVG) method in Section 6 Line 430-431.
>
> **W2 & Q1 & Q2:** Can you more concretely establish the FV similarity is correlated with forgetting, perhaps with a correlation plot?
>
> > Thank you for highlighting this important issue regarding the conclusive correlation between forgetting and FV similarity. We would like to humbly clarify that function vector similarity **indeed statistically coincides** with catastrophic forgetting.
> >
> >
> > Based on the questions in **Q1** and **Q2**, we have now included a correlation plot in the revised paper. The detailed explanation is as follows:
> >
> > - Verify the correlation between FV similarity and forgetting (**Q1**)
> >     - **Correlation plots:** We have provided scatter plots to demonstrate the correlation between model performance and FV similarity. For each test task, we gathered 40 data points from various models (across different task sequences and stages) and plotted correlation diagrams. The results, detailed in **Figure 6 in Appendix F**, show a significant correlation -- as FV similarity decreases, model forgetting increases.
> >      - **Correlation metrics:** We calculated Weighted Kendall's Tau for each plot, achieving strong correlations of 0.645, 0.797, and 0.706 for Hellaswag, Alpaca, and CommonsenseQA, respectively.
> > - Comparison to other similarity metrics (**Q2**)
> >     - **Correlation plots:** In **Figure 6**, we also depict the results concerning the **similarity of the last layer hidden states and the L2 distance of parameters**. Our analysis reveals that the similarity of hidden states does not effectively indicate the presence of forgetting, while the L2 distance exhibits a modest correlation that is **weaker** than the correlation observed with FV similarity.
> >     - **Correlation metrics:** We calculated Weighted Kendall's Tau for L2 distance, yielding scores of 0.444, -0.396, and 0.511 in Hellaswag, Alpaca, and CommonsenseQA, respectively. This further confirms that **FV similarity is a more effective predictor of forgetting**.
> > - Object Count is unconvincing (**W2**)
> >     - In the "Object Counting" scenario, we noted **almost no forgetting** for various training states (as depicted in Figure 6-(I)(d), where data points are close to or exceed the model's initial performance). Consequently, the observation that "FV similarity does not correlate with performance" in this specific context was expected since such a correlation is meaningless in the absence of forgetting. This insight encourages us to delve deeper into the mechanisms of positive transfer in LLMs in the future work

---

> ### Author Response · Authors · 2024-11-23
> **Response to Reviewer ZyxB (2)**
>
> **W3 & Q3 & Q4:** How would you refute the effectiveness of intervention being from returning a specific layer to the behavior of an earlier model? Why FV is a measure of latent task identification?
>
> > We appreciate the reviewer's insightful comments regarding the effectiveness of our function vector framework and the potential alternative explanations.
> >
> >
> > To begin, we would like to humbly emphasize **function vectors are representations of specific tasks for a model**, practically and theoretically, and are capable of influencing the model's entire behavior.
> >
> > Why FV is a measure of latent task identification? (**Q3**)
> >
> > - **Practically**, function vector is indeed effective in regulating the final outputs.
> >     - **Function vector can control the task behavior of pre-trained model**. In Figure 9, experiments with the Llama2-7b-chat model show that inserting or removing function vectors inputs significantly impacts the model’s zero-shot task performance, whereas random vectors have a negligible influence.
> >     - **Function vector can control the task behavior of trained model**. Figure 4 in the main text indeed illustrates that adding source FV or removing target FV effectively mitigates forgetting in trained models, underscoring their causal influence on task behavior.
> >     - **Task specificity of function vectors**. According to interventions in Figure 4, adding FVs from unrelated tasks minimally affects model performance, confirming their task-specific nature.
> > - **Ideally**, the extraction of FV meets the latent variables assumption in LLM as outlined in Section 5 Line 336-352. The main insights are as follows:
> >     - Latent variable assumption for in-context learning: $P_M(y|p,x) = \int_{\Theta} P_M(y \mid \theta, x) P_M\left( \theta \mid p, x\right) d \theta$
> >     - Identifying FVs involves two parts: extract mean activation $\bar{h}$ from inputs $[p, x]$ ($\bar{h}$ equals $\theta$ both conditioned on $p, x$ ); search $\bar{h}$ with highest casual-effect $P_M(y \mid \bar{h}, x)-P_M(y \mid x)$ (equals to find $\theta$ with highest $P_M\left( \theta \mid p, x\right)$). So **identify FVs is to identify the task latent variables** in LLMs.
> >
> > How would you refute the effectiveness of intervention being from returning a specific layer to the behavior of an earlier model? (**Q4**)
> >   - **Model updates are not restricted to specific layers.**. We report the L2 distances of parameters of Llama2-7b-chat across layers before and after tuning on NI-Seq-G1/NI-Seq-M1 in the **following table**. The data indicate that the most significant shifts in the model occur in the middle and later layers. As FV interventions occur before layer 15, it highlights FV's **capacity to influence the entire model**.
> >   - **Function vector is efficient for pre-trained models**. The ability of FVs to manipulate pre-trained models illustrates that such interventions do more than simply restore shifted representations.
> >
> > | NI-Seq-G1 |  |  |  | NI-Seq-M1 |  |  |  |
> > | --- | --- | --- | --- | --- | --- | --- | --- |
> > | Layers | L2 | Layers | L2 | Layers | L2 | Layers | L2 |
> > | 0-3 | 36 | 16-19 | 140 | 0-3 | 64 | 16-19 | 260 |
> > | 4-7 | 38 | 20-23 | 218 | 4-7 | 67 | 20-23 | 406 |
> > | 8-11 | 48 | 24-27 | 166 | 8-11 | 84 | 24-27 | 292 |
> > | 12-15 | 76 | 27-31 | 123 | 12-15 | 130 | 27-31 | 223 |

---

> ### Author Response · Authors · 2024-11-23
> **Response to Reviewer ZyxB (3)**
>
> **W4 & Q5:** How regularization on the FV of particular tasks affect the whole forgetting?
>
> > Thank you for raising this important question regarding the procedure and work mechanism of our function vector-guided training.
> >
> > To clarify, when training on a task, regularization is imposed **based solely on the most recently trained task**. We have now updated Eq.4 and Eq.5 to include the training state for better understanding.
> >
> > Regarding your insightful concern (**Q5**): When training with only the activations for "task 1" being regularized, can the model effectively mitigate forgetting on "task 2" (another unrelated task)? The answer is yes. Below, we elaborate on the `working mechanism` and other `empirical results` of our proposed method that support this design choice.
> >
> >
> > - We have illustrated the `working mechanism` in Figure 5 of our revised submission.
> >   - Suppose that training on task $T_0$ establishes a predictive pathway (shown in orange) that aligns well with the task.
> >   - If no regularization is applied, learning of a new task $T_1$ will necessarily update the function attention heads, i.e., $P_M(\theta|x)$, (shown in red blocks), producing new function vectors $\theta^1_{T_0}$ and $\theta^1_{T_1}$ that are biased toward $T_1$. These shifts in function vectors lead to a derailed predictive pathway (shown in purple) with erroneous predictions for task $T_0$; in other words, forgetting of $T_0$ occurs. In summary, as stated in Line 400, the modifications in $P_M(\theta|x)$ rather than $P_M(y|x,\theta)$ are the primary driving force behind forgetting.
> >   - Our proposed regularization in Eq.4 aims to regularize the distance between $\theta^1_{T_1}$ and $\theta^0_{T_1}$, thus preventing unnecessary modifications to $P_M(\theta|x)$ and mitigating forgetting.
> >   - Our proposed regularization in Eq.5 further limits changes in $\theta^1_{T_1}$, by aligning the predictive probabilities conditioned on the current function vector $\theta^1_{T_1}$ with those of the pre-trained model.
> >   - Together, these two regularization terms ensure minimal changes to $P(\theta|x)$ for each recent task, thereby preserving the performance of previously learned tasks.
> >
> > - We indeed presented some `experimental results`, demonstrating the **effectiveness of function vector guided training to alleviate forgetting on "task 2".**
> >     - In Table 2, our method indeed reduces the decline in both general and in-context learning performance (referred to as "task 2" in this context), underscoring the efficacy of FVG.
> >     - Figure 5 indeed illustrates the **stability** of the "task 2" function vector during FVG training. It demonstrates that FVG **effectively prevents the function vector shift** in unrelated tasks like Hellaswag and CommonsenseQA, thus mitigating forgetting.
> >
> > - We also present some `empirical results` that were conducted even during algorithm development. We considered replacing the current function vector with the previous tasks' in Eq.5. Unfortunately, as shown in the following table, intervening with previous function vectors leads to **even worse performance**. This further justifies our working mechanism detailed above, considering that:
> >    - The previous function vector, e.g., $\theta^0_{T_0}$, is computed under previous tasks (e.g., $T_0$);
> >    - Under the continual learning setup, the model has no access to previous tasks (e.g., $T_0$) when training on the current task (e.g., $T_1$);
> >    - Consequently, forcing the predictive probabilities conditioned on **previous function vectors**, e.g., $\theta^0_{T_0}$, to align with the pre-trained model **at the current task** likely drives even more modifications of $P(\theta|x)$ to fit the current task.
> >
> >
> > |    NI-Seq-G1 on Llama2 | GP | IP | FP |
> > | --- | --- | --- | --- |
> > |  FVG|  50.50  | 56.19 | 22.19  |
> > | FVG with Previous FV | 48.71 | 43.26 | 18.46 |
> >

---

> ### Author Response · Authors · 2024-11-23
> **Response to Reviewer ZyxB (4)**
>
> **Q6:** The effect on plasticity of the proposed method.
>
> > Thank you for your insightful feedback regarding the claim that our training method does not affect plasticity. We appreciate your inquiry into the limitations of this assertion and the potential impacts of our regularization technique on learnability in extreme scenarios.
> >
> >
> > Below, we test on different sizes of datasets to assess the effects of our regularization on plasticity.
> > - Experimental results on small-scale fine-tuning datasets: We tested the model's plasticity on a subset of the training data used in the paper. Specifically, this subset contains 1,000 samples, with the first three tasks being classification tasks and the last three being generative tasks. We observed that for these datasets, **our method sacrifices almost no plasticity.**
> >
> > | Llama2-7b-chat  |NI1510 | NI1343 | NI1292 | NI1290 | NI511 | NI1357 |
> > | --- | --- | --- | --- |--- | --- | --- |
> > |Original fine-tuning| 1.00  | 0.99 | 0.82  | 0.30 | 0.20 |0.28|
> > |Function vector guided training | 0.99 | 0.99 | 0.81 | 0.28 | 0.19 |0.28|
> >
> > - Experimental results on large-scale fine-tuning datasets: We followed your suggestion and tested the model's performance on OpenMathInstru-2 with sample sizes up to 30k due to resource constraints, as shown in the table below. The results presented in the table are trained after 10 epochs. There exists a performance disparity for FVG compared to original fine-tuning. This may be due to slower convergence from regularization. We will continue training the FVG model until full convergence to observe changes in plasticity and will update our new findings promptly. If the issue persists, it indicates limitations in our approach to handling complex tasks, which could guide future research.
> >
> > | OpenMathInstruct-2 on Llama2-7b-chat | 5k| 15k | 30k |
> > | --- | --- | --- |--- |
> > |Original fine-tuning| 0.32  | 0.38 | 0.44  |
> > |Function vector guided training | 0.29 | 0.33 | 0.36 |
> >

---

> > ### Comment · Reviewer_ZyxB · 2024-11-24
> >
> > Thank you for your extremely thorough rebuttal and additional experiments. This is a great paper and your responses have addressed my questions. I have raised my score to a 10 and will advocate for acceptance.

---

> ### Author Response · Authors · 2024-11-25
> **Official Comment by Authors**
>
> We are thrilled to note that your key concerns have been effectively addressed. We sincerely appreciate your dedicated time and effort in examining our paper and offering invaluable and positive feedback.

---

### Official Review · Reviewer_KUE1 · 2024-11-04

**Soundness:** 3
**Presentation:** 2
**Contribution:** 3
**Rating:** 8
**Confidence:** 4

**Summary:**

The paper explores catastrophic forgetting in continual instruction finetuning across various settings, discovering that model forgetting is influenced by the specific training tasks and the models themselves. It explores correlation between function vectors and catastrophic forgetting, and  propose a function vector-based approach to eliminate catastrophic forgetting. The paper has conducted ablation studies across various tasks, methods and models to demonstrate the generalizability of the method using different llama and mixtral models with classification and generation tasks and evaluating it for supervised finetuning and model generalizability.

**Strengths:**

1. The idea of using regularization on the function vectors and the function vector guided loss to enable continual instruction fine tuning seems novel and is well motivated in the paper.
2. The paper presents a strong ablation study for the proposed methods across various settings, demonstrating their generalizability by conducting experiments using multiple llama checkpoints, classification and generation tasks and evaluating across generalizability and supervised finetuning on multiple datasets.
3. The experiment studies which clearly demonstrate the ability of  function vectors to significantly improve the performance on continual instruction fine tuning.

**Weaknesses:**

1. There is a lack of clarity in the proposed methodology. i.e function vector guided training design. It would help to understand  the proposed methodology by providing the overall  the optimisation objective and end-to-end training algorithm. In the current state, it is difficult to reproduce the results that the authors mentioned in the paper without the details on the proposed methodology.
2. The  presentation  of the paper and the readability of some sections are difficult to follow.   Notations are unnecessarily complicated.
3. The experimental results though convincing across most of the experimental setups, its hard to make decisive conclusions especially on the core idea of function vectors and their effectiveness in mitigating catastrophic forgetting.
4. Some experiments and  observations made related to catastrophic forgetting are redundant and may not contribute much. For example, forgetting will be model-dependent since performance on different datasets and tasks is model-dependent.

More details on the above mentioned points are listed below.

1. The authors claim that “Forgetting coincides with changes in FV similarity between tasks”. However, it is difficult to observe the same from the evidence provided because the trend is not constant across all tasks and models; sometimes, a slight change in similarity changes the task performance drastically(i.e. Fig 2(a) on count object and hellaswag datasets), in some cases even though similarity is decreasing, the performance is increasing. It is very difficult to establish a clear relation from the empirical results provided. A suggestion is to  extend the number of tasks or conduct a statistical test using correlation metrics.
2. The observation that generation tasks lead to greater forgetting cannot be measured directly, even though the same evaluation metric (Rouge) is used for both types of tasks. Rouge is more sensitive to longer sequences because it relies on n-grams overlap. The tasks have different difficulties; one way to measure them is by comparing them against some upper bound baseline and calculating forgetting in terms of percentage loss.
3. The computation of the FV guided loss (eqn 4) is not very clear. If the set S consists attention head with top-10 CE,  is that going to be same before training on the current task and after training. Is this loss recomputed at every iteration of the optimization step?
4. Although function vectors improve the performance of existing CL methods, it can be helpful to compare against  finetuning models individually, to see how much gap there is between methods with function vector guided fine tuning and the optimal performance that can be achieved by fine tuning.

**Questions:**

1. It is unclear in methodology of training with function vector guided loss. Since attention head will change at every iteration during training, when do you impose the KL divergence-based regularization. Since, it requires separate computation using ICL-based examples, how do you update model weights through backdrop?  Kindly provide more details on the FV loss computation.
2. The FV-based method may require additional computational resources, especially during interventions where multiple layers are evaluated. Since to extract FV, one has to run the model on a set of counterfactual inputs to isolate activation patterns and then select top attention with the highest causal effect. The FV-guided KL-divergence loss aligns logits produced with and without FV interventions. This requires computing model outputs twice. There needs to be a discussion on the computational time and memory overhead required for the proposed approach.
3. While training on a task, regularization is imposed just based on the most recently trained task or do you consider all previously trained tasks? The loss function in equation 4 and 5 may be modified to consider the state of training on a particular task in the sequence.
4. There are some typos in the paper Line 107: bracket missing in citation, Line 970: it should be We instead of I, Line 231  : sequensce --> sequence.

---

> ### Author Response · Authors · 2024-11-23
> **Response to Reviewer KUE1**
>
> Thank you for your insightful comments and for taking the time to review our paper. Below, we have provided detailed responses to your comments, along with revisions to the manuscript, which are highlighted in blue for your convenience. We hope our responses and revisions adequately address the concerns and provide additional clarity. Please feel free to let us know if you have any additional questions or feedback.
>
> **W1:** There is a lack of clarity in the proposed methodology, especially the optimization objective and end-to-end training algorithm.
>
> > Thank you for your valuable feedback regarding the clarity of our proposed methodology.
> >
> > - In response to your comments, we have revised our **manuscript (Lines 410-429)** to include the overall optimization objective.
> > - Additionally, we have added the end-to-end **training algorithm in Appendix B**, which outlines the extraction of function vectors and the optimization procedure of function vector guided training. These updates aim to provide clarity on how the training process aligns with the optimization objective, thereby facilitating a better understanding of our methodology.
> > - To ensure reproducibility, we have elaborated on the implementation details in **Appendix E.** Additionally, we are committed to releasing our code in the near future.
>
> **W2:** The presentation of the paper and the readability of some sections are difficult to follow.
>
> > We sincerely appreciate the reviewer's thoughtful suggestions, which have guided us in making several revisions to improve the clarity and readability of our paper:
> >
> > - We have now included a detailed description of the **training algorithm** used in our study in Appendix B.
> > - We have added illustrations in Figure 5 and detailed discussions in Section 5 Line 336-352 of **the causal pathway contributing to forgetting**, along with the motivations behind our Function Vector Guided (FVG) method in Section 6 Line 430-431.
> > - We have **revised the figure captions (Figure 1-10)** to highlight our main conclusions and integrate connections to supporting data for a more cohesive presentation.
>
> **W3 & D1:** It is hard to make decisive conclusions on the core idea of function vectors and their effectiveness in mitigating catastrophic forgetting
>
> > Thank you for pointing out this issue. Following your valuable suggestion in **D1**, we would like to humbly clarify that function vectors (FVs) are indeed effective in mitigating catastrophic forgetting.
> >
> > - Function vectors **ideally** mitigate forgetting given that **'Forgetting statistically coincides with changes in FV similarity'**.
> >   - **Correlation plots:** We have included scatter plots to demonstrate the correlation between model performance and FV similarity. For each test task, we gathered 40 data points from various models (across different task sequences and stages) and visualized them in correlation diagrams. The results, detailed in **Figure 6 of Appendix F**, reveal a significant correlation -- as FV similarity decreases, model forgetting increases.
> >   - **Correlation metrics:** We calculated Weighted Kendall's Tau for each plot, achieving strong correlations of 0.645, 0.797, and 0.706 for Hellaswag, Alpaca, and CommonsenseQA, respectively. Note that the low correlation of 0.035 for Ob-count is expected, as there is no observable forgetting in this case, making such a correlation meaningless.
> > - Function vector also **practically** mitigate forgetting.
> >   - In **Table 2**, our method demonstrates its ability to significantly reduce forgetting in general and in-context learning performance. This confirms the practical effectiveness of FVs in mitigating forgetting.
> >   - To further validate the role of FVs, we conducted an ablation study **in the following table**. Compared with (a) the standard KL loss that regularizes parameter changes and (b) KL loss with random vector loss that replaces FV in Eqn. (5) with a random vector, incorporating FVs not only prevents forgetting effectively but also meantime increases plasticity.
> >
> > |     | NI-Sep-G1 |   |   | NI-Sep-C1 |     |     |
> > | --- | --- | --- | --- | --- | --- | --- |
> > |     | GP  | IP  | FP  | GP  | IP  | FP |
> > | IncLora | 47.16 | 30.94 | 19.35 | 45.83 | 27.71 | 83.80 |
> > | (a) KL Loss | 50.99 | 54.55 | 21.46 | 49.39 | 54.75 | 83.90 |
> > | (b) KL with random vector loss | 48.67 | 47.56 | 18.49 | 48.32 | 41.2 | 56.43|
> > | (ours) KL with FV loss | **51.57**| **55.46** | **21.96** | **51.09** | **53.18** | **85.50** |

---

> ### Author Response · Authors · 2024-11-23
> **Response to Reviewer KUE1 (2)**
>
> **W4:** Some experiments and observations are redundant and may not contribute much. For example, forgetting will be model-dependent.
>
> > We would like to humbly clarify the role of observations on model-dependent forgetting, and apologize if there is any misunderstanding here.
> > - The **dependence of forgetting on models**, as shown in Figure 1 is **distinct from the dependence of performance on models**. For instance, while Llama3 is widely recognized for its higher capability and stronger performance, it exhibits more pronounced forgetting on Object-count tasks (see the third column in C-II of Figure 1).
> > - These observations highlight differences in the **specific abilities of models to combat forgetting, which is separate from their problem-solving capabilities**. The severe forgetting exhibited by Llama3 underscores this distinction and offers novel insights that we believe will be valuable to the community.
> > - The primary objective of highlighting the model-dependent nature of forgetting is to **provide a clear motivation for our use of FVs**.
> >    - This model-dependence inspired us to analyze forgetting from an internal model perspective with respect to task awareness, which ultimately guided us to adopt FVs as a means of understanding and mitigating forgetting.
> >    - Such an approach addresses a gap in previous work. For example:
> >       - References [1] and [2] employ feature-based similarity that is agnostic to model specifics, which risks overlooking critical internal model dynamics.
> >       - Reference [3], on the other hand, focuses solely on changes within the model itself, ignoring task-specific details.
> >
> >    Our method bridges these gaps by combining model-specific insights with task awareness.
> >
> > [1] Vinay V. Ramasesh, et.al., Anatomy of Catastrophic Forgetting: Hidden Representations and Task Semantics.
> >
> > [2] Sebastian Lee, et.al., Continual Learning in the Teacher-Student Setup: Impact of Task Similarity.
> >
> > [3] Sen Lin, et.al., Theory on Forgetting and Generalization of Continual Learning.
>
> **D2:** The observation that generation tasks lead to greater forgetting cannot be measured directly. One way to measure them is by comparing them against some upper bound baseline and calculating forgetting in terms of percentage loss.
>
> > Thank you for pointing out that our conclusion regarding the use of Rouge-L to determine that generation task sequences lead to greater forgetting is inappropriate.
> > - In the main text, we reported Rouge-L scores in Table 1 and used it to compare generation and classification task performances in the "Task type perspective" analysis. However, this does not directly support our conclusion, as  Rouge-L  is more sensitive to generation tasks.
> >
> > - A fairer analysis would be to compare percentage loss relative to the upper bound baseline. **We have indeed presented this in Figure 1**, where the numbers on top of each subfigure denote **the testing performance right after training a task**. We take these performance numbers as upper bound baselines, and compute/visualize percentage changes relative to them to reflect forgetting in each subfigure. The results indicate that when training on the NI-seq-C1 sequence, the performance of previously learned tasks **remains around 91%**, while generation tasks may **drop below 30%**. Again conclude that generation task sequences lead to greater forgetting.
>
> **D3 & Q1:** The computation of the FV guided loss (eqn 4) is not very clear.
>
> > - Set $\mathcal{S}$ consists of attention heads with the top-10 CE values. We found that $\mathcal{S}$ changes very slowly, even during training without regularization, as detailed in Figure 10, Appendix F. After completing training on five different tasks, the original Set $\mathcal{S}$ still maintains a high ranking and continues to play a crucial role as an important set of attention heads.
> >
> > - The optimization process for Eq.4 is as follows:
> >
> >   1. We extract the function vector head set $\mathcal{S}$ from a series of held-out datasets; remarkably, $\mathcal{S}$ demonstrates strong generalization across datasets [1] and remains fixed throughout training.
> >
> >   2. When training task $T_j$, we first extract the activation $h_{lk}^{M_{j-1}}$ of attention heads belonging to $\mathcal{S}$ using model $M_{j-1}$ for every sample in training dataset $\mathcal{D}_j$. These activations are saved to memory for further use.
> >
> >   3. At each iteration, for each sample $x$, we extract the activation $h_{lk}^{M}(x)$ of attention heads in $\mathcal{S}$ using the current model $M$. We then compute the loss $\ell_{FV}$ in Eq.4 and optimize the model weights  through backpropagation.
> >
> >     These steps are now clearly detailed in Algorithm 1, Appendix B.
> >
> >
> > [1] Todd Eric, et. al., Function vectors in large language models.

---

> ### Author Response · Authors · 2024-11-23
> **Response to Reviewer KUE1 (3)**
>
> **Q3:** While training on a task, regularization is imposed based on which tasks?
>
> > When training on a task, regularization is imposed **based solely on the most recently trained task**.  Below, we elaborate on the `working mechanism` and other `empirical results` of our proposed method that support this design choice.
> > - We have illustrated the `working mechanism` in Figure 5 of our revised submission.
> >   - Suppose that training on task $T_0$ establishes a predictive pathway (shown in orange) that aligns well with the task.
> >   - If no regularization is applied, learning of a new task $T_1$ will necessarily update the function attention heads, i.e., $P_M(\theta|x)$, (shown in red blocks), producing new function vectors $\theta^1_{T_0}$ and $\theta^1_{T_1}$ that are biased toward $T_1$. These shifts in function vectors lead to a derailed predictive pathway (shown in purple) with erroneous predictions for task $T_0$; in other words, forgetting of $T_0$ occurs. In summary, as stated in Line 400, the modifications in $P_M(\theta|x)$ rather than $P_M(y|x,\theta)$ are the primary driving force behind forgetting.
> >   - Our proposed regularization in Eq.4 aims to regularize the distance between $\theta^1_{T_1}$ and $\theta^0_{T_1}$, thus preventing unnecessary modifications to $P_M(\theta|x)$ and mitigating forgetting.
> >   - Our proposed regularization in Eq.5 further limits changes in $\theta^1_{T_1}$, by aligning the predictive probabilities conditioned on the current function vector $\theta^1_{T_1}$ with those of the pre-trained model.
> >   - Together, these two regularization terms ensure minimal changes to $P(\theta|x)$ for each recent task, thereby preserving the performance of previously learned tasks.
> >  - We indeed present some `experimental results`, demonstrating the **effectiveness of function vector guided training to alleviate forgetting.**
> >     - In Table 2, our method indeed reduces the decline in both general and in-context learning performance, underscoring the efficacy of FVG.
> >     - Figure 5 indeed illustrates the **stability** of the evaluation task function vector during FVG training. It demonstrates that FVG **effectively prevents the function vector shift** in unrelated tasks like Hellaswag and CommonsenseQA, and thus mitigating forgetting.
> >
> > - We also present some `empirical results` that are conducted even during algorithm development. We considered replacing the current function vector with the previous tasks' in Eq.5. Unfortunately, as shown in the following table, intervening with previous function vectors leads to **even worse performance**. This further justifies our working mechanism detailed above, considering that:
> >    - The previous function vector, e.g., $\theta^0_{T_0}$, is computed under previous tasks (e.g., $T_0$);
> >    - Under the continual learning setup, the model has no access to previous tasks (e.g., $T_0$) when training on the current task (e.g., $T_1$);
> >    - Consequently, forcing the predictive probabilities conditioned on **previous function vectors**, e.g., $\theta^0_{T_0}$, to align with the pre-trained model **at the current task** likely drives even more modifications of $P(\theta|x)$ to fit the current task.
> >
> >
> > |    NI-Seq-G1 on Llama2 | GP | IP | FP |
> > | --- | --- | --- | --- |
> > |  FVG|  50.50  | 56.19 | 22.19  |
> > | FVG with Previous FV | 48.71 | 43.26 | 18.46 |

---

> ### Author Response · Authors · 2024-11-23
> **## Response to Reviewer KUE1 (4)**
>
> **D4**: How much gap there is between methods with function vector guided fine-tuning and the original fine-tuning?
>
> > In the table below, we present the best $FP$ performance of FVG on different datasets using the Llama2 model, comparing it with the performance of individual finetuning for each task. The results indicate that FVG outperforms individual fine-tuning on NI-Seq-G1, likely due to enhanced inter-task knowledge transfer. However, for the Trace sequences, a noticeable gap remains between FVG and direct fine-tuning.
> >
> > It is also important to emphasize that in this paper, FVG **focuses on addressing the forgetting of existing knowledge ($GP$ and $IP$)** in the model. While other continual learning algorithms may achieve better performance on the FP metric, our method is specifically designed to enhance GP and IP metrics without compromising the model's ability to retain existing knowledge. As shown in Table 2, FVG effectively achieves this balance, demonstrating strong performance in preserving prior knowledge while still adapting to new tasks.
> >
> > | | NI-Seq-G1 | NI-Seq-C1 | NI-Seq-M1 | Trace |
> > | --- | --- | --- | --- | --- |
> > | Individual Fine-tuning | 25.44 | 85.90 | 61.21 | 60.30 |
> > | Best-of-FVG | 28.05 | 84.50 | 58.61 | 53.87 |
>
>
> **Q2:** There needs to be a discussion on the computational time and memory overhead required for the proposed approach.
>
> > As discussed in the response to D3 & Q1, along with the detailed algorithm procedure provided in Algorithm 1, Appendix II, we now outline the computation and memory overhead associated with FVG.
> >
> > - **Calculating set $\mathcal{S}$**: This is performed only once for the entire training sequence.
> > - **Calculating function vector $\theta$**: This is computed once for each task.
> > - **Calculating the attention head activation for FV consistency loss**: This is computed once for each task and stored with minimal memory overhead (approximately 10 KB per sample).
> > - **Calculating the intervention logits FV-guided KL-divergence loss**: This is performed once during each iteration.
> >
> >As a result, the primary overhead in practice arises from the additional computation required for the KL-divergence loss, as it necessitates calculating model outputs twice. However, this **forward pass overhead is minor** compared to the computational cost of the backward pass and remains acceptable within practical training scenarios.
> >
> >To quantify this, we measured the time required for 100 iterations under both fine-tuning and FVG training. The results, summarized in the table below, demonstrate that the additional cost introduced by FVG is minimal:
> >
> > |     | Fine-tuning | FVG training |
> > | --- | --- | --- |
> > | Time cost for 100 iter |  109 second   |  149 second   |
>
> **Q4:** There are some typos in the paper
>
> > Thank you very much for your careful review and for pointing out the typos in our paper. We have corrected these typos in the revised version and will thoroughly polish the paper further.

---

> ### Author Response · Authors · 2024-11-25
> **Official Comment by Authors**
>
> We sincerely appreciate your dedicated effort in reviewing our manuscript. As the rebuttal is coming to a close, we kindly remind you that we have submitted a response to your comments. We would be grateful if you could confirm whether our responses have addressed your concerns? If you have any additional concerns, please don't hesitate to let us know.

---

> > ### Comment · Reviewer_KUE1 · 2024-11-26
> > **Reply to Author Comments**
> >
> > Thank you for the detailed rebuttal, which addressed several questions regarding the paper and provided greater clarity to the proposed approach.

---

> > > ### Author Response · Authors · 2024-11-27
> > > **Official Comment by Authors**
> > >
> > > We are thrilled to note that your key concerns have been effectively addressed. We sincerely appreciate your dedicated time and effort in examining our paper and offering invaluable and positive feedback.

---

### Official Review · Reviewer_G6Jm · 2024-11-04

**Soundness:** 3
**Presentation:** 3
**Contribution:** 3
**Rating:** 8
**Confidence:** 3

**Summary:**

The paper investigates Catastrophic Forgetting in Large Language Models during continual learning, revealing that forgetting is influenced by specific tasks and model characteristics, and proposes a novel training methodology that utilizes a function vector to stabilize performance and reduce CF, with empirical validation across multiple benchmarks.

**Strengths:**

* The forgetting issue during fine-tuning foundation models is an important problem.
* The observations and analysis through the function vector in this paper is very interesting, which I believe would be interesting to many.
*  The finding that CF in LLMs primarily stems from biases in function activation rather than the overwriting of task processing functions is not obvious, but I think the authors has done a good job in confiming this.
* The writing is easy to follow.

**Weaknesses:**

Overall, I like the findings in the paper and the authers have done extensive investigations. I am interested in comparing the proposed method with model averaging, which is shown in [1] that model averaging is very effectivenss in terms of mitigating forgetting. Furthermore, is it possible to analyze the effectivenss of model averaging through function vector?


[1] Yong Lin, et.al., Mitigating the alignment tax of RLHF.

**Questions:**

See above

---

> ### Author Response · Authors · 2024-11-23
> **Response to Reviewer G6Jm**
>
> Thank you for your insightful comments on our paper. Please kindly find our response to your comments below. Additionally, all modifications to the manuscript have been highlighted in blue for easy reference. We hope that our responses and revisions adequately address the concerns you've raised. Please feel free to let us know if you have any additional concerns or questions.
>
> **W1:** Comparison between the proposed method and model averaging [1] method in mitigating forgetting.
>
>
> > Thank you for suggesting a comparison with Model Averaging, one of the state-of-the-art methods in mitigating forgetting. This will enable us to demonstrate the effectiveness of function vector in characterizing and alleviating forgetting from multiple perspectives.
> >
> > - We **evaluate Model Averaging** [1] on NI-Sep-G1/NI-Sep-C1 with a combination of IncLora/EWC methods on Llama2/Llama3 models. A shortcut of the results is provided **in the following table** while the entire results can be found in Table 8 in Appendix F.
> > - We perform model averaging on the pre-trained model and the final model with the averaging ratio set to 0.2, according to [1]. It shows **better performance** on fine-tuned datasets (**FP**) but **struggles in** the general/in-context learning evaluation setting (**GP/IP**) compared to function vector guided training.
> >
> >We have incorporated these comparisons and discussions in Appendix F.
> >
> > | NI-Sep-G1 on Llama2 | GP  | IP  | FP  |
> > | --- | --- | --- | --- |
> > | IncLora | 47.16 | 30.94 | 19.35 |
> > | + Model averaging | +0.87 | +10.35 | +1.46 |
> > | + FVG | **+3.34**	|**+25.25**|**+2.84**|
> > | EWC | 33.48 | 26.87 | 17.72 |
> > | + Model averaging | +6.58 | +11.27 | **+2.59** |
> > | + FVG | **+15.73** | **+27.18** | +0.85 |
> >
> > [1] Yong Lin, et.al., Mitigating the alignment tax of RLHF.
>
> **W2:** Analysis on the effectiveness of model averaging through function vector?
>
> > - While model averaging helps in reducing forgetting, it's intriguing to examine this through the lens of the function vector. We present the changes in the function vector before and after model averaging, along with the corresponding performance data, in **Figure 8 Appendix F.**
> > - The results show a **significant correlation between performance and FV similarity in model averaging methods**. Compared to its final model provider, IncLora, model averaging reduces FV shifts and improves performance. Additionally, an analysis across different training stages reaffirms the positive correlation between performance and FV shifts.

---

> > ### Comment · Reviewer_G6Jm · 2024-11-27
> >
> > A follow up question is that in this new experiment you set a fixed averaging ratio set to 0.2. It is hard to compare Model averaging and FVG since one is better at the fine-tuned dataset but the other is better at general/in-context learning evaluation setting. It is better to change the averaging ratio to plot the trade-off curve and compare these two methods.

---

> ### Author Response · Authors · 2024-11-27
> **With Sincere Gratitude for Your Review, We Humbly Request Your Confirmation on Our Responses**
>
> We sincerely appreciate your dedicated effort in reviewing our manuscript. As the rebuttal is coming to a close, we kindly remind you that we have submitted a response to your comments. We would be grateful if you could confirm whether our responses have addressed your concerns? If you have any additional concerns, please don't hesitate to let us know.

---

> > ### Comment · Reviewer_G6Jm · 2024-11-27
> >
> > Thank you for addressing my questions. I don't have any further concerns and raise the scores accordingly

---

> ### Author Response · Authors · 2024-12-03
>
> We are thrilled to note that your key concerns have been effectively addressed. We sincerely appreciate your dedicated time and effort in examining our paper and offering invaluable and positive feedback.
>
> We have conducted additional experiments using various averaging ratios (0.2, 0.4, 0.6, 0.8), and the results are presented in the table below. The experiments demonstrate that model averaging (MA) exhibits better plasticity in our scenario, while the FVG proposed in this paper has a stronger advantage in mitigating forgetting.
>
> | Llama2 on NI-Seq-G1|	GP |	IP |	FP|
> | --- | --- | --- | --- |
> |IncLora |	47.16	| 30.94	| 19.35 |
> |+MA_0.2 |	48.03 |	41.29	| 20.81 |
> |+MA_0.4 |	49.79 |	48.76 |	**23.26** |
> |+MA_0.6	| 49.15 |	54.25 |	23.51 |
> |+MA_0.8	| 49.23 |	55.16 |	19.45 |
> |+FVG	| **50.50** |	**56.19** |	22.19 |

---

### Author Response · Authors · 2024-11-23
**General Response**

Dear Reviewers and ACs,

We sincerely thank all the reviewers and ACs for your diligent efforts and high-quality reviews. If you have any additional questions or require further clarification, please feel free to let us know. Your insights are highly valued.

We are delighted to note that reviewers find that:

- Our hypothesis and method are novel, insightful (Reviewers `KUE1`, `Qiar`), interesting (Reviewers `G6Jm`, `ZyxB`), well-motivated (Reviewer `KUE1`), thorough (Reviewers `Qiar`, `ZyxB`), and presented in a clear and easy-to-follow writing style (Reviewers `G6Jm`, `Qiar`).
- Our observations and analysis on forgetting through the function vector are very interesting (Reviewers `G6Jm`, `ZyxB`), strong, thorough (Reviewers `ZyxB`, `KUE1`), and have significantly contributed to the field (Reviewer `Qiar`).
- Our paper studies the forgetting problem, which is important (Reviewer `G6Jm`). It concludes that CF in LLMs primarily stems from biases in function activation rather than the overwriting of task processing functions, a finding that is well-received (Reviewer `G6Jm`) and insightful (Reviewers `Qiar`, `ZyxB`). It also provides an effective approach based on the function vector to mitigate forgetting (Reviewer `KUE1`).

In response to your valuable suggestions, we have conducted additional experiments and made the following modifications in the Rebuttal-PDF for your convenience:

- Table 1: We updated the results for NI-Seq-G1 on Llama2-7b-chat to correct a previous setting error in the hyperparameters.
- Figure 3: We added intervention results with FV from an unrelated task (suggested by Reviewer `ZyxB`).
- Figure 4: We included a comparison with the original fine-tuning process.
- Figure 5: Illustration of the causal pathway to forgetting (suggested by Reviewers `KUE1`, `ZyxB`).
- Figure 6: Correlation plot on model performance and different similarity metrics (suggested by Reviewers `KUE1`, `ZyxB`).
- Table 8, Figure 8: Analysis and performance of Model Averaging (suggested by Reviewer `G6Jm`).
- Figure 9: Intervention results on four datasets via function vector on pre-trained models (suggested by Reviewer `ZyxB`).
- Algorithm 1: Procedure for function vector-guided continual learning (suggested by Reviewer `KUE1`).
- Loss function: In lines 410-429, we updated the loss with more concrete notations (suggested by Reviewer `KUE1`).
- Captions for Figures 1-10: We updated the captions to highlight the 'Main conclusion' along with its substantial supporting data (suggested by Reviewer `ZyxB`).
- Related work: We added a one-sentence summary to connect the related works with our study (suggested by Reviewer `Qiar`).
- Notation shape: In lines 127-130, we clarified the shape of the notation used in our paper (suggested by Reviewer `Qiar`).
- Due to space limitations, we moved Table 1 and Figure 3 in the old version of the paper to Table 5 and Figure 7 in the Appendix.

Best regards,

The Authors

---

### Meta-Review · Area_Chair_ss76 · 2024-12-22

**Metareview:**

The authors study the use of Function Vectors (Todd et al., 2023) in the context of catastrophic forgetting. They show that:
1) Task similarity, as measured by the cosine distance between the respective function vectors of each task, is correlated with the amount of forgetting;
2) By intervening on the trained model using function vectors of previous tasks, they can mitigate the performance drop due to forgetting;
3) By training the model with additional regularization that depends on the function vectors of previous tasks, they can also mitigate forgetting without test-time intervention. The high-level idea is that function vectors can be kept around as compact representations of previous tasks (instead of keeping all of the previous training data around), and used for regularization to avoid forgetting.
Overall, reviewers found the study and proposed method exciting, and unanimously recommended acceptance. I recommend acceptance as well and I agree that this paper will be interesting and useful to the community!

There were two broad concerns raised. First, several reviewers pointed out that the writing was terse and hard to follow, with various grammatical errors, typos, and missing details. The authors addressed this partially in their revision, though the revised version still contains several writing issues (especially when introducing the details of Function Vectors in Section 2.2 and the evaluations in Section 3). Second, several reviewers pointed out places where the conclusions the authors drew were not rigorously supported by the experiments (e.g., in Section 4, which covers the correlation between function vector similarity and forgeting). The authors responded well to these comments by adding more experiments to the appendix. I encourage the authors to significantly update their camera-ready version to incorporate the reviewers' feedback, make the paper clearer to readers unfamiliar with function vectors and prior work on catastrophic forgetting, and reorganize the paper such that the new results are also represented in the main text. I think this will help the broader community appreciate and build on this good work.

As a final suggestion, I might have missed it, but I didn't see an ablation of the two components of the new proposed loss (equations 4 and 5). That, and the comparison to model averaging that was in the rebuttal, will help us better understand what about the proposed regularization works well

**Additional Comments On Reviewer Discussion:**

There were two broad concerns raised. First, several reviewers pointed out that the writing was terse and hard to follow, with various grammatical errors, typos, and missing details. The authors addressed this partially in their revision, though the revised version still contains several writing issues (especially when introducing the details of Function Vectors in Section 2.2 and the evaluations in Section 3). Second, several reviewers pointed out places where the conclusions the authors drew were not rigorously supported by the experiments (e.g., in Section 4, which covers the correlation between function vector similarity and forgeting). The authors responded well to these comments by adding more experiments to the appendix.

---

### Decision · Program_Chairs · 2025-01-22

Accept (Oral)